# Pre-activation of autophagy impacts response to olaparib in prostate cancer cells

Maxime Cahuzac [1,2], Patricia Langlois [1,2], Benjamin Péant[1,2], Hubert Fleury[1,2], Anne-Marie Mes-Masson [1,2,3✉] & Fred Saad [1,2,3]

Poly (ADP-ribose) polymerase 1 (PARP1) plays an essential role in DNA repair and is targeted by anticancer therapies using PARP inhibitors (PARPi) such as olaparib. PARPi treatment in prostate cancer (PC) is currently used as a monotherapy or in combination with standard therapies (hormonotherapy) in clinical trials for patients with DNA damage response mutation. Unfortunately, 20% of these patients did not respond to this new treatment. This resistance mechanism in PC is still not well understood. Here, we report that autophagy affects differently the response of PC cell lines to olaparib depending on its activation status. Pre-activation of autophagy before olaparib resulted in an increase of DNA repair activity by homologous recombination (HR) to repair double-strand breaks induced by olaparib and enhanced cell proliferation. When autophagy was activated after olaparib treatment, or completely inhibited, PC cells demonstrated an increased sensitivity to this PARPi. This autophagy-mediated resistance is, in part, regulated by the nuclear localization of seques-trosome 1 (SQSTM1/p62). Decrease of SQSTM1/p62 nuclear localization due to autophagy pre-activation leads to an increase of filamin A (FLNA) protein expression and BRCA1/Rad51 recruitment involved in the HR pathway. Our results reveal that autophagy basal levels may in part determine amenability to PARPi treatment.

[1] Centre de recherche du Centre hospitalier de l'Université de Montréal (CRCHUM), Montreal, QC, Canada. [2] Institut du cancer de Montréal, Montreal, QC, Canada. [3] Department of Surgery, Université de Montréal, Montreal, QC, Canada. ✉email: anne-marie.mes-masson@umontreal.ca

Prostate cancer (PC) is the most frequently diagnosed cancer for North American men, with >260,000 new cases each year[1,2]. Early-stage prostate cancer has an excellent prognosis with excellent 5- and 10-year overall survival with local therapy ± androgen deprivation therapy (ADT). In advanced prostate cancer, especially when patients become resistant to ADT (known as castration-resistant prostate cancer or CRPC), the available therapeutic options are non-curative and survival is generally <3 years. Therapeutic options include taxane-based chemotherapy and more recently novel hormonal therapies, that directly or indirectly target the androgen receptor (AR), such as abiraterone and enzalutamide. Eventually, patients develop resistance to available therapeutic options and succumb to their disease. Ongoing research continues to better understand and develop therapeutic approaches in patients who fail novel hormone therapies. One avenue of intense research in this area is in the use of PARP inhibitors in prostate cancer.

Germline or somatic mutations in DNA damage response (DDR) genes, particularly in homologous recombination (HR)[3–5], are found in up to 30% of metastatic PC cases. These mutations can be targeted with new anticancer therapies using poly (ADP-ribose) polymerase 1 (PARP1) inhibitors (PARPi)[6–8] such as olaparib. PARPi inhibit the PARP1/2 enzymes involved in DNA damage repair[9] and induce synthetic lethality in cancer cells that have deficiencies in HR-mediated DNA repair, such as breast and ovarian cancers that harbor *BRCA* mutations[8]. Currently, PARPi are under investigation for PC as a monotherapy or as a combination treatment with standard therapies such as hormonotherapy, chemotherapy, and radiotherapy[10,11]. One of these clinical trials had shown that olaparib treatment in patients with DDR gene mutations who were also treated with abiraterone had an improved progression-free survival compared to men treated with abiraterone alone (13.8 months vs. 8.2 months)[4,12]. While different mechanisms of resistance to PARPi have been reported for ovarian and breast cancers, as BRCA1/2 reverse mutation or induction of the senescence phenotype[13–16], PARPi resistance in PC remains poorly understood. Recently, autophagy has emerged as a mechanism of multidrug resistance for various therapies in PC[17–19] and may be implicated in resistance to PARPi.

Autophagy is a regulated process that recycles proteins and organelles in the cell and maintains cellular homeostasis under stress conditions, such as nutrient deprivation or oxidative stress. Degraded organelles and proteins are encased in a double-membrane structure called the phagophore, which matures into an autophagosome, also called the autophagic vacuole (AV) that fuses with a lysosome to form an autolysosome (AL) to generate amino acids. Sequestrosome 1 (SQSTM1/p62) is one of the proteins that is degraded during the autophagic process. This ubiquitin-binding protein targets other proteins for degradation through autophagy or proteasomal pathways[20]. Previous studies have shown that SQSTM1/p62 is a mediator that links autophagy to DNA repair, particularly with HR[21,22]. In addition, SQSTM1/p62 has been used as a marker of autophagy and aberrant levels of SQSTM1/p62 have been associated with PC and cancer progression[23–25]. Several studies on PC have also reported autophagy as a pro-oncogenic mechanism through the upregulation of autophagy genes by the AR pathway or C/EBPß[26,27]. Importantly, autophagy has been targeted for potential combinatory therapies in PC to bypass resistance to hormone/chemotherapies[28–30], and previous studies have described that autophagy could affect the response to PARPi in ovarian and breast cancers[31,32]. However, the role of autophagy in PARPi resistance in PC is currently unknown.

In the present study, we investigated the impact of autophagy on PC cell response to the PARPi olaparib by following the autophagy activation timeline. We show that autophagy must be activated before olaparib treatment for a cytoprotective effect. This is regulated in part by the localization of SQSTM1/p62 in the nucleus and its role in the proteasomal degradation of the filamin A (FLNA), an actin-binding protein that interacts with DDR proteins. Pre-activation of autophagy by rapamycin before olaparib treatment reduces the level of SQSTM1/p62 in the nucleus and increases the capacity of PC cells to repair the DNA damage induced by olaparib. Using CRISPR/Cas9 to generate a knockout (KO) of the autophagy-related gene *Atg16L1*, we confirm these results and demonstrate that autophagy pre-activation affects olaparib sensitivity of PC cells and may contribute to PARPi resistance.

## Results

**PC cell lines show different profiles of olaparib sensitivity and basal levels of autophagy.** To investigate the possible link between autophagy and the response to PARPi in PC, we first determined the IC$_{50}$ values for olaparib for three PC cell lines: LNCaP and C4-2B are AR-positive, and PC-3 is AR-negative. Based on dose-response curves to olaparib (Fig. 1a), LNCaP and C4-2B are sensitive (0.17 μM and 0.076 μM, respectively) and PC-3 is resistant (2.13 μM) (Fig. 1b). Next, we examined the PC cell expression of autophagy proteins such as autophagy-related genes (Atg), Beclin-1, and light chain-3 (LC3) A/B, I and II (Fig. 1c and Supplementary Fig. 10a). We observed that the expression of Atg5 and the lipidated form of LC3 A/B, LC3-II, were higher in olaparib-resistant PC-3. Between olaparib-sensitive cells, LNCaP had lower levels of LC3-II than C4-2B. Based on the expression of LC3-II, which represents active autophagy, PC-3 cells seemed to have a higher basal level of autophagy compared to LNCaP and C4-2B. To confirm this, we transduced our cell lines with the Premo$^{TM}$ Autophagy Tandem Sensor construct containing the LC3-II protein tagged with red and green fluorescence protein (RFP/GFP) (Fig. 1d, e). This system allows us to quantify the autophagic flux (ratio of number of AL to AV) in PC cells due to differences in pH sensitivity between RFP and GFP, and track the progression from AV to AL (disappearance of GFP signal due to acidic pH of AL). Treatments with 1 nM of the autophagy activator rapamycin (Rapa) for 24 h alone or in combination with 300 nM of the autophagy inhibitor bafilomycin A1 (Baf) were used as controls. For all cell lines, Rapa treatment significantly increased the number of AL (red puncta) by ~1.5 to 2-fold compared to AV (yellow puncta). Combination with Baf decreased this ratio for LNCaP (4.5-fold change), C4-2B (3-fold change), and PC-3 (6-fold change) (Fig. 1e) cells. These results highlighted that PC-3 demonstrated a higher basal level of autophagy compared to LNCaP and C4-2B (AL/AV ratio = 6.2 compared to 2.5 and 2.7, respectively; Fig. 1e). These observations showed a correlation between basal levels of autophagy in PC cells and olaparib resistance.

**Complete depletion of autophagy increases sensitivity to olaparib.** To determine the role of autophagy levels in olaparib sensitivity, we generated an *Atg16L1* knockout (KO) in LNCaP, C4-2B, and PC-3 cell lines using the CRISPR/Cas9 method. Atg16L1 is an important protein for the lipidation of LC3 A/B[33]. The efficiency of these KO cell lines was confirmed by western blot (Fig. 1f and Supplementary Fig. 10b). Expression of Atg16L1 was completely abolished and lipidation of LC3 A/B (LC3-II) was blocked in all KO cell lines, even in autophagy induction and inhibition by Rapa and Baf treatment. Complete depletion of autophagy reduced the olaparib IC$_{50}$ values to 0.059 μM ($p = 0.036$), 0.009 μM ($p = 0.004$), and 1.5 μM ($p = 0.067$) for LNCaP, C4-2B, and PC-3, respectively (Fig. 1g, h, Table 1). To ensure that PARP1 or PARylation did not affect olaparib

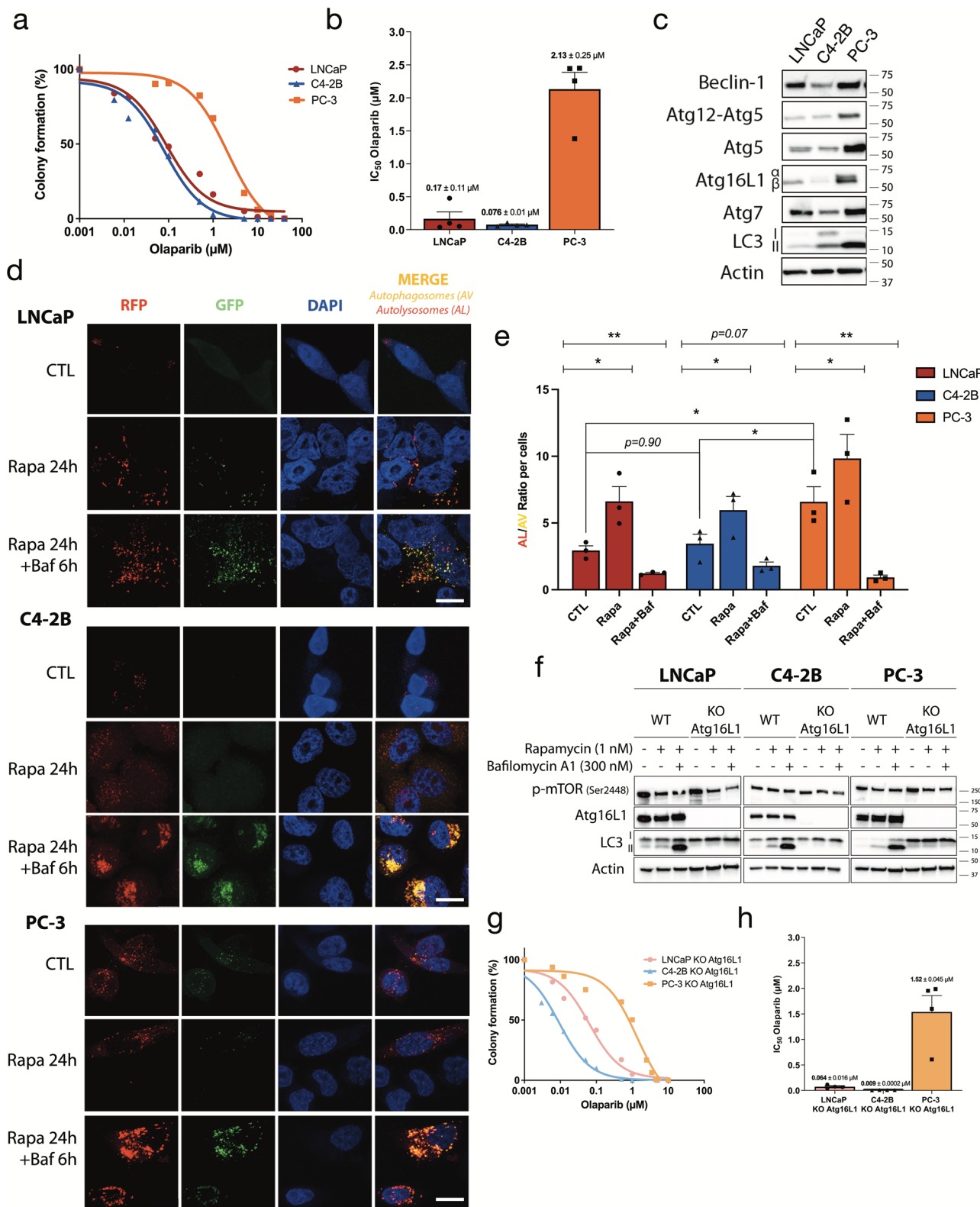

sensitivity, we measured basal level protein by western blot and found no differences between WT PC-3 cells and those undergoing autophagy depletion (Supplementary Fig. 1). Thus, autophagy depletion rendered all PC cells more sensitive to olaparib.

**Pre-activation of autophagy by rapamycin affects the olaparib response of PC cells.** To confirm that a higher level of autophagy mediates a resistant phenotype to olaparib in PC cells, we induced

autophagy with rapamycin 24 h before (denoted as RO10) or 24 h after (denoted as O10R) the start of a 6-day olaparib treatment (Fig. 2a). Rapamycin alone or in combination with olaparib decreased mammalian target of rapamycin complex (mTORC) phosphorylation while increasing LC3-II in all wild-type (WT) cell lines at day 2 and 6 (Fig. 2b, c, Supplementary Figs. 2a, b and 10c respectively). No differences were observed for LC3-II expression in RO10 and O10R conditions indicating similar

**Fig. 1 Basal level of autophagy affects olaparib sensitivity profile of PC cell lines. a** Olaparib sensitivity curves in PC cell lines determined by clonogenic assay. **b** Olaparib $IC_{50}$ of PC cell lines calculated from **a**. **c** Western blot analysis of basal level expression of key autophagy proteins in PC cell lines. **d** Representative images of autophagy flux with the tandem RFP-GFP LC3B sensor in cell lines, captured by confocal microscopy. Rapamycin (Rapa; 1 nM) was used as positive control and combined with 300 nM bafilomycin A1 (Baf) as a negative control. **e** Quantification of autophagy flux was achieved with a specific macro for Image J to calculate the ratio between autolysosomes (AL; red puncta) and autophagosomes (AV; yellow puncta) relative to control. **f** Confirmation of KO *Atg16L1* in PC cell lines under control, autophagy-induced, and autophagy-blocked conditions by western blot analysis. **g** Olaparib sensitivity curves in KO *Atg16L1* cell lines determined by clonogenic assay. **h** Olaparib $IC_{50}$ of KO *Atg16L1* cell lines calculated from **g**. Bars represent average ± SEM of $IC_{50}$ values obtained by three independent clonogenic assays for **b** and **h**. The mean ± SEM of four (**b** and **h**) or three (**e**) independent experiments is shown. Data were analyzed using the two-tail Student *t*-test. *$p < 0.05$ and **$p < 0.01$. Scale bar 10 µm.

**Table 1 $IC_{50}$ values for olaparib between PC wild-type and KO *Atg16L1* cell lines.**

| Cell lines | Olaparib (µM) | | *p-value* |
| | WT | KO Atg16L1 | |
|---|---|---|---|
| LNCaP | 0.17 ± 0.11 | 0.059 ± 0.016 | *p = 0.036* |
| C4-2B | 0.079 ± 0.01 | 0.009 ± 0.0002 | *p = 0.0044* |
| PC-3 | 2.12 ± 0.25 | 1.52 ± 0.045 | *p = 0.067* |

levels of autophagy activation (Fig. 2b and Supplementary Figs. 2b and 10c). Using the IncuCyte live-cell imaging system, we followed the proliferation of LNCaP, C4-2B, and PC-3 cells, which was significantly increased under RO10 conditions (21–50%, *p = 0.021*; 19–56%, *p = 0.0026*; and 38–69%, *p = 0.041*; respectively) after 6 days of culture compared to olaparib treatment alone (O10) (Fig. 2d). This increase was not observed under O10R conditions. When autophagy was completely abrogated in PC KO *Atg16L1* cell lines, proliferation was not significantly up-regulated in RO10 conditions (LNCaP KO *p = 0.35*; C4-2B KO *p = 0.085*; and PC-3 KO *p = 0.058*). We also rescued the depletion of autophagy by introducing a plasmid coding for Atg16L1 with an HA-tag in our PC WT and KO Atg16L1 cell lines (Supplementary Fig. 3). Expression of Atg16L1-HA restored autophagy dynamics by the lipidation of LC3-I in LC3-II that was not observed in PC KO cell lines in autophagy induction and inhibition conditions (Supplementary Fig. 3a and Fig. 1f). Expression of Atg16L1-HA had no effects on autophagy in WT cell lines. PC KO-rescue Atg16L1 also harbored a higher cell proliferation when autophagy was pre-activated (RO10) compared to PC KO Atg16L1 (LNCaP, 48% vs. 23%; C4-2B, 39% vs. 19%; and PC-3, 72% vs. 24%) (Supplementary Fig. 3b and Fig. 2d).

We further examined these results by cell cycle analyses after 2 and 6 days of olaparib treatment (Supplementary Fig. 2c, d and Fig. 2e, respectively). In C4-2B cells, after 6 days of O10 treatment, 70% of cells were blocked in the S phase compared to 7% of control cells as it was shown in different cell lines in the literature[34–36]. Consequently, we observed a decrease in G1 subpopulation from 83% (control) to 9% (O10). Interestingly, when autophagy was activated after O10 treatment (O10R) a similar result was observed with an accumulation of C4-2B cells in the S phase (66%). When autophagy was pre-activated before olaparib treatment (RO10) the proportion of cells in the S phase decreased to 15% and led to an increase of cells in G1 and G2/M (47% and 37%, respectively), indicating that C4-2B cells pre-activated by rapamycin were more proliferative. In LNCaP and PC-3 cells, O10 treatment blocked more cells in the G2/M phase compared to C4-2B (50 and 44%, respectively, vs. 21%) as it was also shown in the literature[16]. We also observed an increase of the sub-G1 phase population in LNCaP and PC-3 cells by ~10%. As observed for C4-2B, RO10 treatment increased the G1 phase population compared to O10 (LNCaP: 64% vs. 26%, and PC-3:

43% vs. 28%), whereas O10R did not change this population. Sub-G1 cells were also decreased in RO10. Moreover, when autophagy was completely abrogated in C4-2B KO *Atg16L1* cells, RO10 treatment did not increase the G1 phase population as observed in WT cells and was similar to the O10 condition, which was ~15%. In contrast, RO10 increased G1 phase cells for LNCaP KO *Atg16L1* cells but to a lesser degree than LNCaP WT cells (*p = 0.03* vs *p = 0.001*). The same result as WT was observed for O10R condition with no significant differences with O10. For PC-3 cells, KO *Atg16L1* cells harbored a different phenotype compared to WT: O10 did not increase the G2/M phase population as observed with WT cells (10% vs. 44%), and RO10 treatment increased the G1 phase population by 1.2-fold. This observation suggests that autophagy depletion acts differently in PC-3 cells compared to other PC cell lines. Thus, activation of autophagy before olaparib treatment affects the PARPi response of LNCaP, C4-2B, and PC-3 by limiting effect of olaparib on cell proliferation and cycle. Interestingly, this protective effect was abrogated when autophagy is activated after olaparib treatment or when it was abrogated by CRISPR/Cas9 and rescued when autophagy was restored.

**Pre-activation of autophagy reduced olaparib-induced DNA double-strand break damage by increasing HR efficiency.** Since olaparib acts by inducing high DNA damage level which induces cell death, we hypothesized that autophagy supports cell proliferation and cell cycle progression through the upregulation of DNA repair to counteract this effect of PARPi. Therefore, we investigated if autophagy activation by rapamycin affects the formation and resolution of H2A histone family member X (γ-H2AX) foci, which can be used as a marker of DNA damage level[37]. WT cells in RO10 treatment had lower levels of γ-H2AX foci per cell after 2 days compared to cells in O10 treatment (LNCaP, 17 vs. 35, *p = 0.0092*; C4-2B, 15 vs. 20, *p = 0.078*; and PC-3, 13 vs. 25, *p = 0.049*) (Fig. 3a and Supplementary Fig. 5a). Although the number of foci decreased after 6 days of treatment, we observed the same difference between RO10 and O10 (Supplementary Fig. 4a). When autophagy was completely abrogated, the number of γ-H2AX was not statistically different between RO10 and O10 conditions for C4-2B and PC-3 KO *Atg16L1* cell lines. LNCaP KO *Atg16L1* cells demonstrated a decrease in γ-H2AX foci that was not as statistically significant as compared to LNCaP WT cells (*p = 0.024* vs. *p = 0.0092*) at day 2. In O10R conditions, levels of γ-H2AX foci were similar from O10 conditions for all cell lines (Supplementary Figs. 4b and Fig. 5a).

As olaparib sensitivity is often related to HR-deficiency and olaparib-induced DNA double-strand break repair, we next measured Rad51 and BRCA1 foci in our cell lines to determine if the decrease of γ-H2AX foci was due to an increase in HR activity (Fig. 3b, Supplementary Figs. 4c and 5b). Autophagy pre-activation (RO10) significantly increased the number of Rad51 foci compared to O10 in all WT cell lines (LNCaP *p = 0.023*; C4-2B *p = 0.045*; and PC-3 *p = 0.031*) (Fig. 3b). Similar to

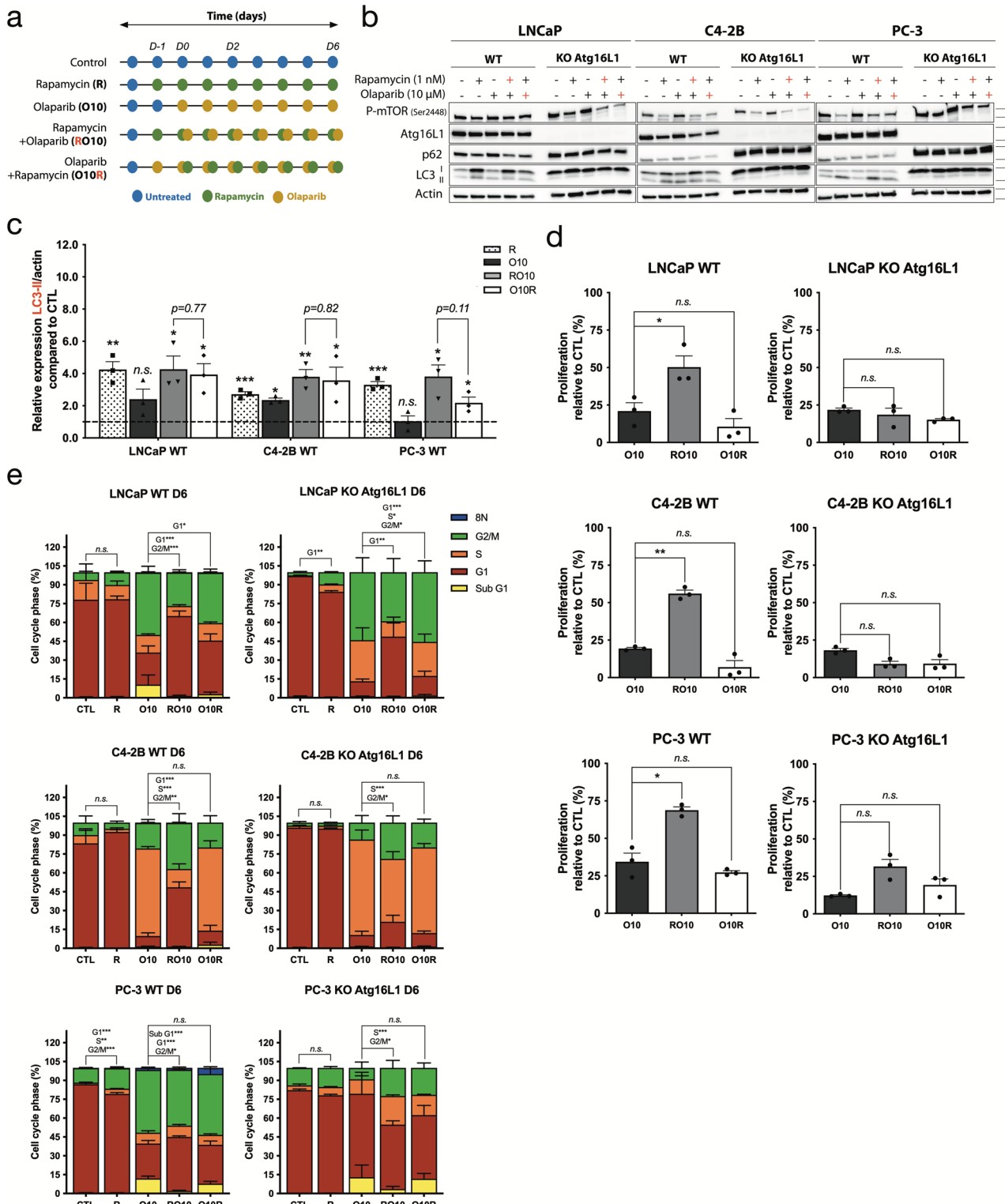

**Fig. 2 Pre-activation of autophagy by rapamycin induces a protective effect against olaparib. a** Treatment timeline of cell lines. Cells were treated with 10 µM olaparib alone (denoted as O10) or received 1 nM rapamycin 24 h before (RO10) or 24 h after (O10R) the start of olaparib treatment. Experiments were conducted for 6 days. **b** Western blot analyses of autophagy induction after olaparib and rapamycin treatments in WT and KO *Atg16L1* cell lines at day 2. Rapamycin (red +) denotes RO10. Olaparib (red +) denotes O10R. **c** Relative expression of LC3-II normalized with actin and compared to control (CTL) from **b**. **d** Cell proliferation of WT and KO cell lines under O10, RO10, or O10R treatments at day 6. **e** Quantification of cell cycle phase populations determined by flow cytometry following 6 days of treatment of WT and KO cell lines. For all data, the mean ± SEM of three independent experiments is shown. Data were analyzed using the two-tail Student *t*-test. n.s. = nonsignificant. *$p < 0.05$, **$p < 0.01$, and ***$p < 0.001$.

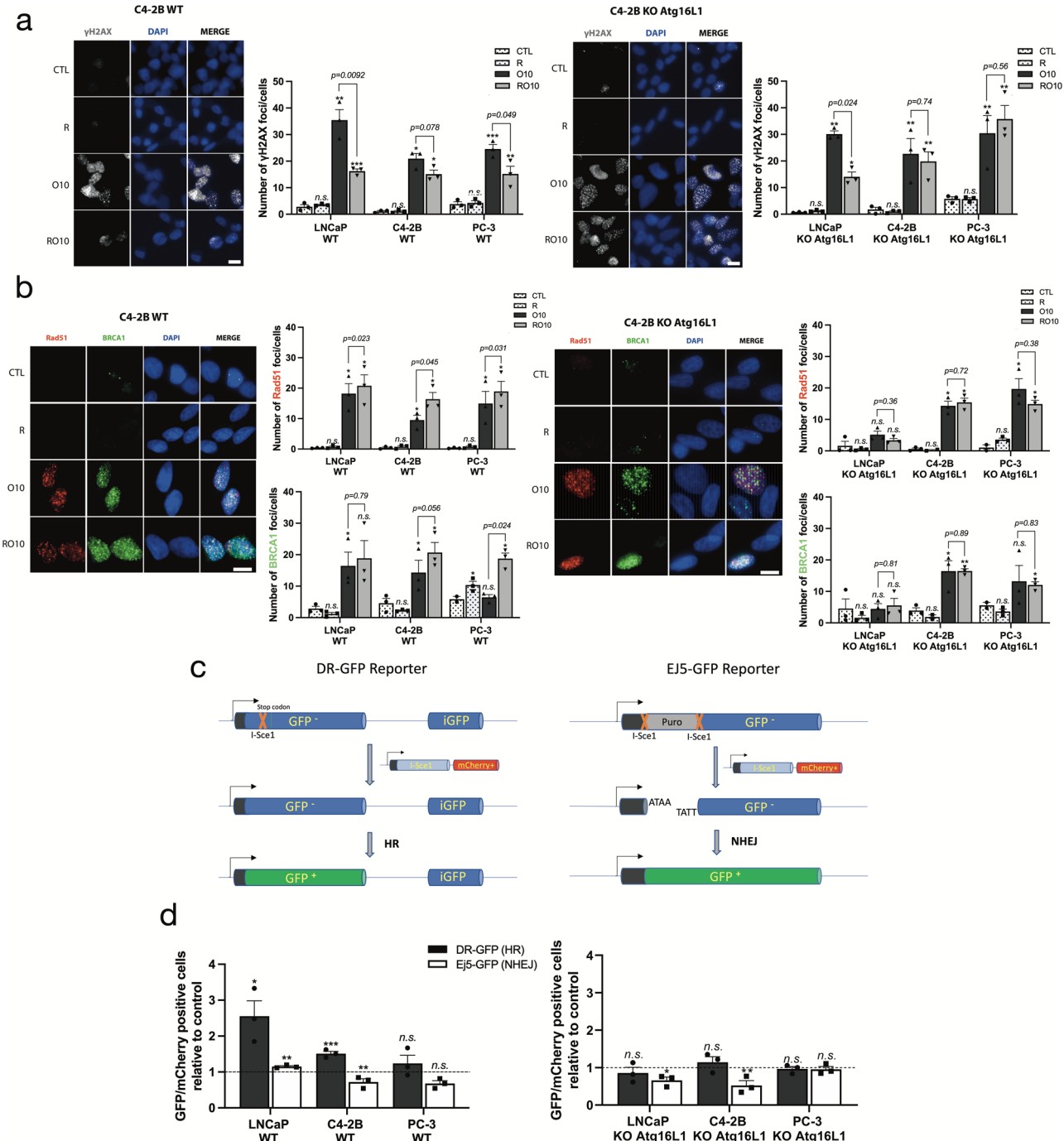

**Fig. 3 Pre-activation of autophagy increases HR activity and reduces the accumulation of DNA damage induced by olaparib. a** Representative images (C4-2B WT and KO *Atg16L1*) and quantification of the number of γ-H2AX foci per nucleus in PC WT and KO cell lines following treatment with rapamycin alone (R), O10 or RO10 after 2 days. **b** Representative images (C4-2B WT and KO *Atg16L1*) and quantification of Rad51 and BRCA1 foci following same conditions as **a**. **c** Schematic representation of reporter assay system. **d** Quantification of HR (DR-GFP) and NHEJ (Ej5-GFP) activity when autophagy was activated by rapamycin 24 h before plasmids co-transfection (RT). The ratio of GFP-positive cells versus mCherry-positive cells was determined by flow cytometry. For all data, the mean ± SEM of three independent experiments is shown. Data were analyzed using the two-tail Student *t*-test. n.s. = nonsignificant. *p < 0.05, **p < 0.01, and ***p < 0.001. Scale bar 10 μm.

γ-H2AX foci, we did not observe a difference between O10 and RO10 in KO *Atg16L1* cells. However, O10R induced lower Rad51 recruitment compared to O10 or RO10 conditions for C4-2B and PC-3 cell lines (Supplementary Figs. 4c and 5b). BRCA1 foci showed a similar pattern to Rad51: PC cells in RO10 conditions had a higher number of BRCA1 foci compared to O10 or O10R. To determine if the RO10-related increase in Rad51-BRCA1

recruitment was representative of DNA repair activity, the function and activity of HR (DR-GFP) and non-homologous end joining (NHEJ) (Ej5-GFP) were independently assessed using a plasmid-based DNA repair GFP reporter assay that allows the quantification of DNA repair by measuring the ligation rate of digested I-SceI ends (Fig. 3c, d and Supplementary Fig. 4d, e). In this assay, GFP-positive cells represent a surrogate for HR and

NHEJ activity. Activation of autophagy with rapamycin before or after DR-GFP/I-Sce1-mCherry or Ej5-GFP/ I-Sce1-mCherry transfection was denoted as RT and TR, respectively. Our results showed that rapamycin treatment prior to the DNA damage induced by the plasmid (RT) significantly increased GFP-positive cells by 2.5-fold for LNCaP ($p = 0.022$) and 1.5-fold for C4-2B ($p = 0.0010$) compared to the control, indicating an increase in HR activity (Fig. 3d). This increase was not significant ($p = 0.35$) for PC-3. In KO cell lines, an increase in HR activity under RT conditions was not observed. The efficiency of NHEJ appeared to decrease in C4-2B WT and KO cells, whereas no difference in NHEJ was observed in LNCaP and PC-3. Under TR conditions, no changes in HR activity were observed for C4-2B, but a slight decrease was observed for LNCaP and PC-3 (Supplementary Fig. 4d, e). Overall, autophagy activation before induction of DNA breaks by olaparib appears to enhance the cell's ability to efficiently repair DNA damage using the HR pathway and not NHEJ.

**Complete depletion of autophagy regulates HR by reducing Rad51/BRCA1 recruitment on DNA damage sites.** To understand the impact of autophagy on PARPi DNA repair response, we characterized the DDR in the WT and KO *Atg16L1* cell lines. Cells were first irradiated with 8 Gy. We followed γ-H2AX foci resolution (Fig. 4a, b and Supplementary Fig. 6a) and Rad51/ BRCA1 recruitment after 30 min and 8-, 24- and 48-h after irradiation (Fig. 4c–e and Supplementary Fig. 6b) by immunocytochemistry. C4-2B and PC-3 KO cell lines had an ~2-fold higher level of γ-H2AX foci compared to WT after irradiation, with 16.5 foci/ WT cell versus 23 foci/ KO cells and 9.2 versus 19.2, respectively (Fig. 4a, b and Supplementary Fig. 6a). LNCaP KO cells also showed higher levels but were not statistically different from LNCaP WT, at 14 foci/cell versus 20. The number of γ-H2AX foci decreased over time, indicating that cells could repair their DNA damage. Interestingly, KO cell lines had less Rad51 and BRCA1 recruitment on DNA damage sites after irradiation compared to WT cells (Fig. 4c–e and Supplementary Fig. 6b). This suggests that HR may be deregulated by autophagy depletion. To confirm this, we used our plasmid-based DNA repair reporter assays, DR-GFP and Ej5-GFP as previously (Fig. 4f). Complete depletion of autophagy significantly reduced the efficiency of HR in C4-2B ($p = 0.048$) and PC-3 ($p = 0.0002$) cell lines. A non-significant decrease ($p = 0.50$) was also observed in LNCaP. Rescued of autophagy using Atg16L1-HA plasmid significantly restored the loss of HR efficiency in PC KO *Atg16L1* cell lines (LNCaP KO-rescue, $p = 0.0036$; C4-2B KO-rescue, $p = 0.02$; and PC-3 KO-rescue, $p = 0.0007$) (Supplementary Fig. 7). Interestingly, NHEJ efficiency remained the same, suggesting that PC KO cell lines preferentially used this pathway to repair their DNA breaks. These results suggest that autophagy impacts HR-mediated DNA repair via BRCA1 and Rad51 recruitment.

**SQSTM1/p62 mediates autophagy-regulated response to olaparib through expression of FLNA.** Since SQSTM1/p62 is considered a key mediator between autophagy and DNA repair[21,38], we examined if SQSTM1/p62 contributed to olaparib resistance when autophagy was pre-activated in PC cell lines. As expected, we observed that the PC KO cell lines had higher expression levels of SQSTM1/p62 compared to WT cells (Fig. 2b). Since nuclear localization of SQSTM1/p62 is important in DNA repair[21,22,38], we used confocal microscopy to determine and quantify the localization of SQSTM1/p62 in our PC WT and KO cell lines (Fig. 5a, b and Supplementary Fig. 8a, b). Olaparib alone (O10) in C4-2B WT cells induced a significant increase

($p = 0.045$) of SQSTM1/p62 puncta in the nucleus (12/cell) compared to control (5/cell) (Fig. 5a, b). In LNCaP and PC-3 WT cells, differences between the control and O10 conditions were not significant (Fig. 5a, b and Supplementary Fig. 8a, b). Interestingly, autophagy pre-activation (RO10) significantly reduced SQSTM1/p62 foci in the nucleus compared to O10 in LNCaP cells (3.58 versus 5.86, $p = 0.045$), C4-2B (6.08 versus 11.16, $p = 0.013$), and PC-3 (5.96 versus 9.46, $p = 0.046$). This decrease was not observed in O10R conditions (Supplementary Fig. 8b). When autophagy was completely depleted in KO cell lines, we did not observe any significant decrease of SQSTM1/p62 in RO10 compared to O10 conditions (Fig. 5b). As observed for γ-H2AX foci, PC-3 KO cells had a higher number of SQSTM1/ p62 foci after olaparib treatment compared to WT (19.8 vs. 10.2). SQSTM1/p62 interacts with and promotes the proteasomal degradation of FLNA protein, which recruits BRCA1 and Rad51 protein at DNA break sites[21]. We determined whether FLNA was important in this autophagy-mediated resistance to olaparib by measuring the nuclear fraction of FLNA and SQSTM1/p62 (Fig. 5c, d and Supplementary Figs. 8c and 10d). RO10 and O10R conditions were compared with O10 as we studied the impact of autophagy pre- and post-activation on FLNA and SQSTM1/p62 nuclear localization after an olaparib treatment. Under autophagy pre-activation (RO10), FLNA was more highly localized to the nucleus in LNCaP WT (2.3-fold change), C4-2B WT (3.7-fold change) and PC-3 WT (3.4-fold change) compared to O10 (Fig. 5c, d), which correlates well with increased Rad51/BRCA1 foci. In contrast, O10R conditions or KO cell lines did not show this increase in FLNA expression under RO10 conditions. This increase of FLNA was accompanied by a decrease of SQSTM1/ p62 in the nucleus in LNCaP WT (3.3-fold change), C4-2B WT (3.4-fold change), and PC-3 WT (4.5-fold change) in RO10 conditions compared to O10 (Fig. 5c and e). This effect was not observed in the O10R condition and in PC KO *Atg16L1* cell lines. These results suggest that a variation in SQSTM1/p62 nuclear localization can affect the levels of nuclear FLNA which in turn may affect BRCA1/Rad51 recruitment during HR-mediated DNA repair.

**Targeting SQSTM1/p62 rescued effect of autophagy in PC KO Atg16L1 cell lines.** To confirm the importance of SQSTM1/p62 in this autophagy-mediated resistance, we used a siRNA against SQSTM1/p62 in our PC KO *Atg16L1* cell lines and in WT ones (Fig. 6 and Supplementary Fig. 9). We followed the sequence of treatment as rapamycin, by transfecting siRNA (si) or scramble (Sble) 24 h before or after olaparib treatment (Sble/siO10, O10Sble/si; respectively) (Supplementary Fig. 9a). We confirmed siRNA efficacity by western blot and observed an important decrease of SQSTM1/sip62 protein level mainly in PC KO *Atg16L1* but also in PC WT cell lines at day 2 and 6 (Fig. 6a and Supplementary Fig. 10e). Interestingly, pre-inhibition of SQSTM1/p62 (siO10) in LNCaP, C4-2B, and PC-3 KO reverses effects of autophagy depletion on cell proliferation after olaparib treatment (15–37%, 16–52%, and 12–38%, respectively), where no differences were previously observed in RO10 conditions (Figs. 2d and 6b). A decrease of SQSTM1/p62 had a similar effect as autophagy pre-activation by rapamycin (RO10), an increase of cell proliferation compared to O10 by 1.7 to 2-fold change for WT cell lines. This phenotype was lost when inhibition of SQSTM1/p62 was performed after olaparib treatment (O10si). No significant differences were also observed in SbleO10 and O10Sble conditions (Supplementary Fig. 9b). Pre-inhibition of SQSTM1/p62 in KO *Atg16L1* cell lines also decreased the number of γ-H2AX foci per cell after 2 days compared to cells in O10 treatment (LNCaP KO, 8 vs. 22, $p = 0.008$; C4-2B KO, 10 vs. 21,

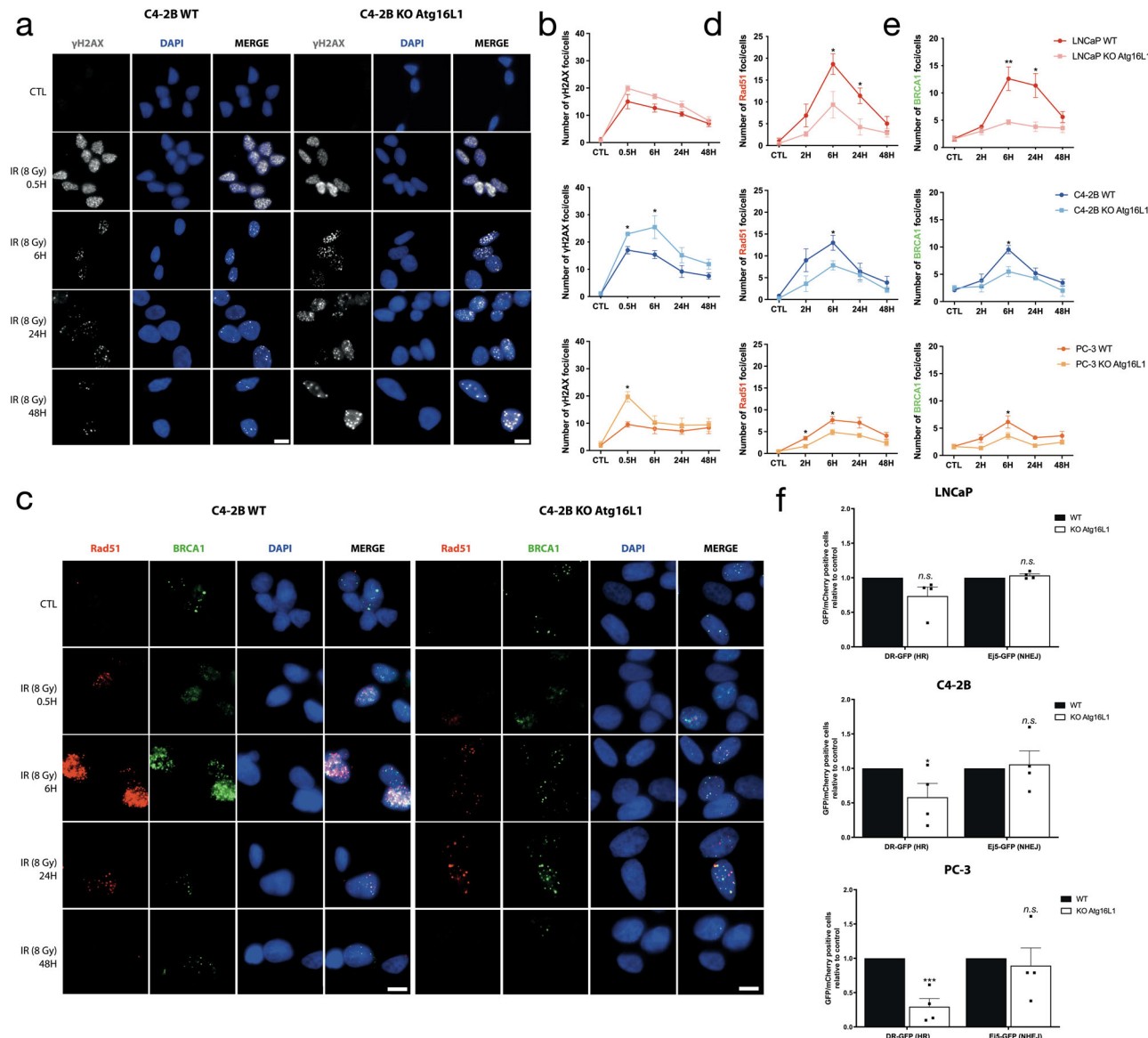

**Fig. 4 Autophagy-depleted PC cells repair DNA breaks using NHEJ. a** Representative images (C4-2B WT and KO *Atg16L1*) of the number of γ-H2AX foci per nucleus in WT and KO cells of C4-2B after 8 Gy irradiation. **b** Quantification of γ-H2AX foci per cells. **c** Representative images (C4-2B WT and KO *Atg16L1*) of the number of Rad51 and BRCA1 foci per nucleus in WT and KO cells of C4-2B after 8 Gy irradiation. **d, e** Quantification of Rad51 and BRCA1 foci per cell. **f** Analysis of HR (DR-GFP) and NHEJ (Ej5-GFP) activity in WT and KO cell lines by flow cytometry. The mean ± SEM of three (**b**) or four (**d**, **e**) independent experiments is shown. Data were analyzed using the two-tail Student *t*-test. n.s. = nonsignificant. *$p < 0.05$,**$p < 0.01$ and ***$p < 0.001$. Scale bar 10 μm.

$p = 0.027$; and PC-3 KO, 11 vs. 18, $p = 0.008$) (Fig. 6c). A similar decrease was observed in WT cells. In all Sble and O10si conditions, levels of γ-H2AX foci were similar from O10 conditions for all cell lines (Supplementary Fig. 9c). To determine if this decrease in γ-H2AX foci was due to an increase in HR efficiency, we used our GFP reporter assay (Fig. 6d). As expected, HR was more efficient in KO *Atg16L1* and WT PC cell lines where SQSTM1/p62 was pre-inhibited (siO10) and not O10si conditions compared to SbleO10 and O10Sble, respectively. This provides evidence that the regulation of SQSTM1/p62 drives the autophagy-mediated resistance observed when autophagy was pre-activated in PC WT cell lines.

## Discussion

Autophagy is considered as a mechanism of multidrug resistance in neuroblastoma and PC and has been targeted to increase the

efficiency of PARPi and chemotherapies, respectively[28,39–41]. However, the role of autophagy in PARPi resistance remains poorly understood in PC. Here, we propose that pre-activation of autophagy before olaparib treatment provided a cytoprotective effect in PC cells that supported proliferation and DNA repair and may provide insights into mechanisms of PARPi resistance (Fig. 6e). When autophagy is pre-activated (higher basal level) before olaparib treatment, nuclear localization of SQSTM1/p62 is reduced leading to higher expression of FLNA by limiting its proteasomal degradation. This permits cells to have a more efficient HR and that could contribute to a resistant phenotype to olaparib. If autophagy is activated after treatment or completely inhibited, FLNA expression is decreased by SQSTM1/p62 levels, HR is less functional and PC cells are sensitized to olaparib.

Several studies have focused on the effect of autophagy depletion to increase the efficiency of conventional

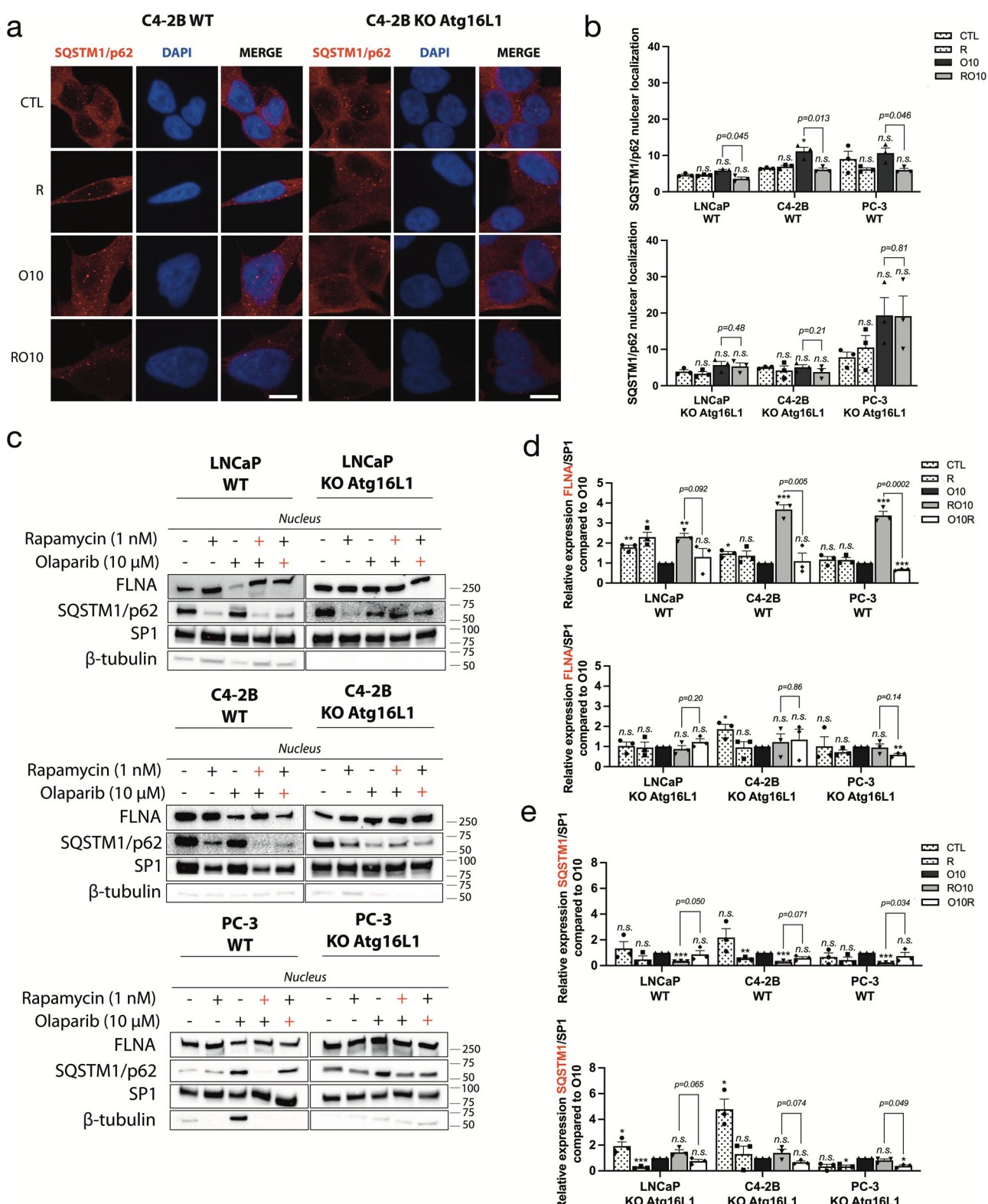

**Fig. 5 SQSTM1/p62 nuclear localization regulates autophagy-mediated resistance to olaparib. a** Representative images (C4-2B WT and KO *Atg16L1*) of the number of SQSTM1/p62 puncta following rapamycin alone (R), O10 or RO10 treatments after 2 days. **b** Quantification of nuclear SQSTM1/p62 in WT and KO cell lines. **c** Western blot of nuclear fraction after 2 days of RO10 and O10R treatment in PC WT and KO *Atg16L1* cell lines. Rapamycin (red +) denotes RO10. Olaparib (red +) denotes O10R. ß-tubulin was used as quality control as a marker of cytoplasmic fraction and SP1 as a marker of the nuclear fraction. **d**, **e** Relative expression of FLNA and SQSTM1 normalized with SP1 and compared to O10 condition. For all data, the mean ± SEM of three independent experiments is shown. Data were analyzed using the two-tail Student *t*-test. n.s. = nonsignificant. \**p* < 0.05, \*\**p* < 0.01, and \*\*\**p* < 0.001. Scale bar 10 μm.

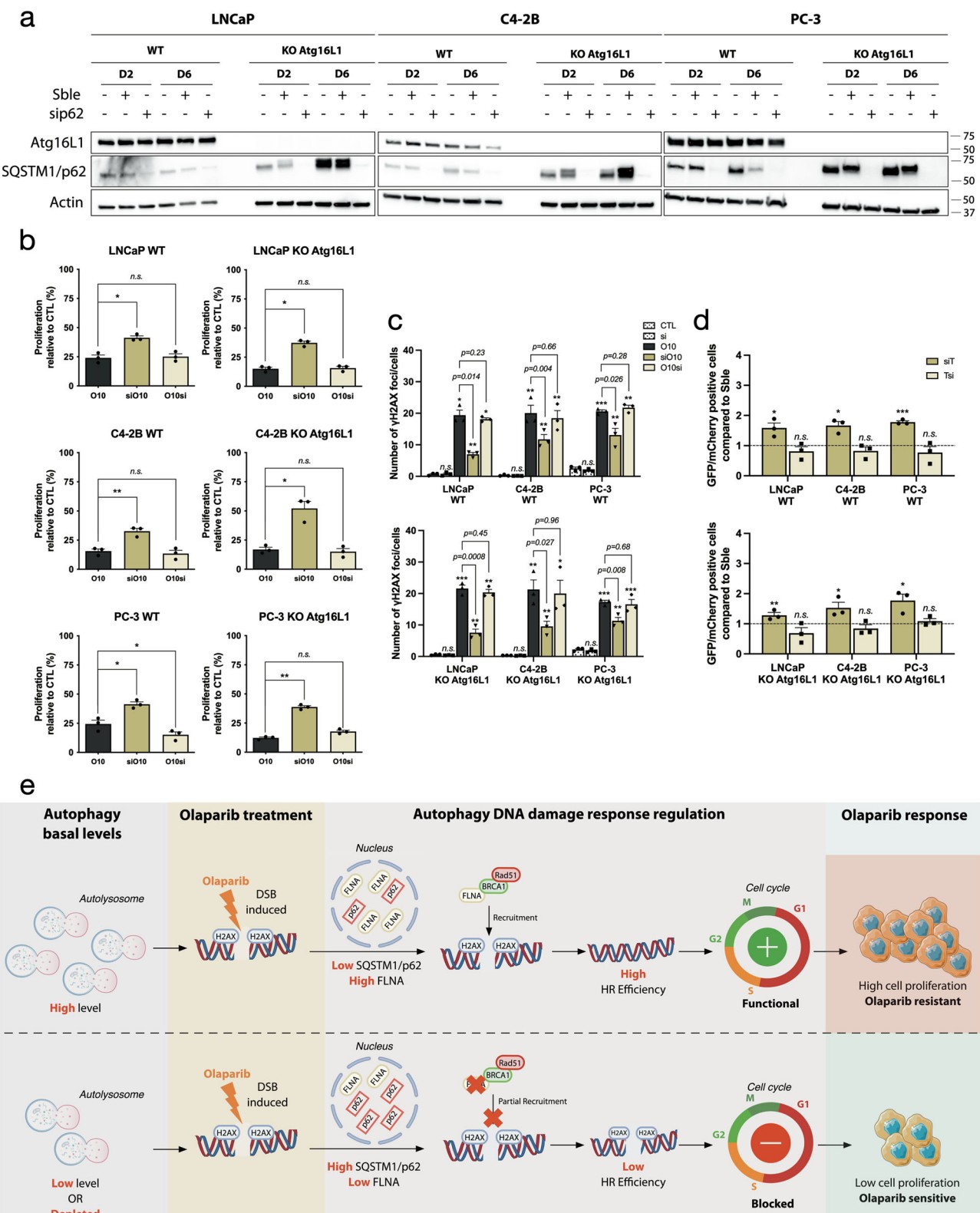

treatments (hormono- and chemotherapies) in PC[19,28,29,42]. We observed a similar effect in which autophagy depletion significantly increased PC sensitivity to olaparib and showed that the basal activation level of autophagy impacts this sensitivity. PC-3 cells had higher basal levels of autophagy than the other PC cell lines and were more resistant to olaparib. This higher basal level of autophagy is consistent

with reports that describe autophagy enhances prostate tumor growth[26].

In our study, we decided to investigate whether the sequencing of autophagy activation with rapamycin before or after olaparib treatment would affect PARPi response. When autophagy was activated after olaparib treatment (O10R), we observed a slight decrease in cell proliferation (Fig. 2d). This result was not

**Fig. 6 Targeting SQSTM1/p62 rescue autophagy-mediated resistance in PC KO *Atg16L1* cell lines. a** Validation of efficacity of siRNA against SQSTM1/p62 at day 2 and 6. **b** Cell proliferation of PC WT and KO cell lines following transfection with siRNA against SQSTM1/p62 alone (si) or 24 h before olaparib treatment (siO10) or after (O10si) at day 6. **c** Quantification of the number of γ-H2AX foci per nucleus in PC WT and KO cell lines in same conditions as **b. d** Quantification of HR (DR-GFP) activity when SQSTM1/p62 was targeted with a siRNA 24 h before (siT) or after (Tsi) plasmids co-transfection. For all data, the mean ± SEM of three independent experiments is shown. Data were analyzed using the two-tail Student *t*-test. n.s. = nonsignificant. *$p < 0.05$, **$p < 0.01$, and ***$p < 0.001$. **e** This schematic proposes how autophagy activation may determine the PC cell response to olaparib. When a higher level of autophagy is present prior to olaparib treatment, nuclear levels of SQSTM1/p62 are low that prevents degradation of FLNA thereby promoting HR by recruiting Rad51 and BRCA1 to DNA breaks induced by olaparib. This allows PC cells to proliferate, leading to a resistance phenotype. If autophagy level is low prior to olaparib treatment, SQSTM1/p62 is not sufficiently cleared leading to higher levels of SQSTM1/p62 to target FLNA for degradation. This downregulation of FLNA reduces the efficiency of HR, leading to partial repair of DNA breaks and an olaparib-sensitive phenotype. Parts of this schematic proposes were drawn using pictures from Servier Medical Art. Servier Medical Art by Servier is licensed under a Creative Commons Attribution 3.0 Unported License.

surprising because it was shown in the literature that rapamycin and olaparib co-treatment reduced cell proliferation, invasion, and tumor progression in lung and prostate cancers[43–45]. However, when autophagy is activated by rapamycin prior to olaparib treatment (RO10), autophagy protects PC WT or PC KO-rescue *Atg16L1* cells against this PARPi and not PC KO *Atg16L1* ones. These results indicate the importance of the basal level of autophagy and the timing of its activation on the PARPi response.

Autophagy is involved in the DNA repair of double-strand breaks by acting on KAP1 (KRAB (Kruppel-Associated Box Domain)-Associated Protein 1)/STAT3 or SQSTM1/p62 pathways, which regulate HR by interacting with BRCA1/2 and FLNA, respectively[21,46]. Here, we focused mainly on SQSTM1/p62 because of its accumulation in our KO PC cell line models. Autophagy pre-activation by rapamycin leads to increased HR activity and resolved more efficiently γ-H2AX foci. We observed this pattern in all three cell lines at different levels that likely reflected differences in their AR expression, autophagy basal levels, DNA repair efficiency or even the different degrees of autophagy activation by rapamycin[47–49]. Interestingly, we observed a similar level of γ-H2AX foci after 2 and 6 days of treatment between WT and KO cell lines despite the decreased HR efficiency with autophagy depletion. We highlighted that PC KO cell lines repaired DNA breaks using the NHEJ system instead of HR (Fig. 4 and Supplementary Fig. 6), which would explain how γ-H2AX foci were resolved or decreased even during low Rad51 and BRCA1 recruitment and complete depletion of autophagy. Alternatively, autophagy may also affect pathways other than DNA repair. Autophagy may affect olaparib availability after it enters in the cell as drugs can be trapped in lysosomes and undergo degradation during the autophagy process or exocytosis, depending on their basic pKa ≈8 or the expression of P-glycoprotein 1 (P-gp1) at the surface of lysosomes[50–53]. Because olaparib has a basic pKa = 0.2, P-gp1 may reduce the PARPi availability in cells and permit olaparib accumulation in lysosomes, leading to its degradation.

Autophagy has been considered an ideal target to increase treatment efficacy in various cancers. In PC, seven clinical trials have used an autophagy inhibitor in combination with hormono- or chemotherapies but most have not demonstrated efficacy[54,55]. Our results suggest that the autophagy basal level in cancer cells and/or timing of sequential treatment may impact treatment outcomes and efficacy. Moreover, measuring the basal autophagy level of patient tumors/tissues in ex vivo tumor models by using a combination of autophagy markers as SQSTM1/p62 and Atgs may help determine a patient's response to PARPi[56,57].

In summary, this study reveals that autophagy can contribute to the PC cell response to olaparib by regulating SQSTM1/p62 nuclear localization and DNA repair efficiency and provides a potential mechanism of PARPi resistance. Evaluating autophagy basal levels in patient tissues may help determine which patients will be responsive or amenable to PARPi therapy and provides considerations in improving the efficacy of combination therapies in PC using autophagy inhibitors, particularly their sequence/timing in their treatment course when combined with PARPi therapy.

## Methods

**Cell culture.** Human PC cell lines, LNCaP and PC-3, were purchased from the American Type Culture Collection (ATCC CRL-174, ATCC CRL-250, respectively). C4-2B cells were gifted by Dr. Gleave (Vancouver Prostate Centre). All cell lines were maintained in RPMI 1640 (Wisent Inc., 350-000-EL) supplemented with 10% fetal bovine serum (FBS), 0.5 ug/mL amphotericin B (Wisent Inc., 450-105-QL), and 50 ug/mL gentamicin (Life Technologies Inc., 15710064). All PC cell lines were authenticated in 2019 using short tandem repeat (STR) profiling by the McGill University Genome Center (Montreal, Canada). All cell lines were tested negative for mycoplasma with IDEXX BioAnalytics (Columbia, MO65201). To generate KO *Atg16L1* for each cell line, the CRISPR/Cas9 method was used to target exon 1. Cells were transfected using Lipofectamine 3000 (Thermo Fisher Scientific, L3000-015) with two RNA guides against the *Atg16L1* sequence (5′ AAACCCGCTGGAAGCGCCACATCTC 3′) gifted by Dr. Russel (University of Ottawa, ON, Canada). Selection with puromycin (Invitrogen, ant-pr-1) was used 48 h after transfection and was maintained for at least 2 weeks. Limiting dilutions were performed for clonal selection. KO *Atg16L1* cell lines were verified by western blot and short tandem repeat (STR) DNA profiling was performed. Rescue experiment was performed using a plasmid coding for Atg16L1 with a HA-tag gifted by Dr. Russel. siRNA against SQSTM1/p62 were purchased from Horizon Discovery (J-010230-05-0020 and J-010230-07-0020).

**Reagents and antibodies.** Olaparib (Selleckchem, AZD2281), rapamycin (LKT Labs, 53123-88-9), and bafilomycin (Sigma-Aldrich, 88899-55-2) were used. The following antibodies were used for the study: Beta-Actin (AC14) (abcam, AB6276, 1:20000 dilution); Atg16L1 (D6D5) (Cell Signaling, 8089 T, 1:1000 dilution); Atg5 (D5F5U) (Cell Signaling, 12994 S, 1:1000 dilution); Atg12 (D88H11) (Cell Signaling, 4180 S, 1:1000 dilution); Beclin-1 (D40C5) (Cell Signaling, 3495 S, 1:1000 dilution); Atg7 (D12B11) (Cell Signaling, 8558 S, 1:1000 dilution); β-Tubulin (D2N5G) (Cell Signaling, 15115 S, 1:1000 dilution); Filamin A (Cell Signaling, 4762 S, 1:1000 dilution); SP1 (Sigma, PLA0307, 1:5000 dilution); anti-phospho-Histone H2A.X (Ser139) (Sigma-Aldrich, JBW301, 1:2000 dilution); LC3 A/B (D3U4C) (Cell Signaling, 12741 S, 1:750 dilution); phospho-mTOR (Ser2448) (D9C2) (Cell Signaling, 5536 T, 1:1000 dilution); SQSTM1/p62 (Cell Signaling, 5114 T, 1:1000 dilution) and (abcam, ab56416, 1:100 dilution); Rad51 (114B4) (abcam, ab213, 1:750 dilution) and BRCA1 (Sigma Millipore, 07-434, 1:1000 dilution), PARP1 (proteintech, 66250, 1:650 dilution) and PAR/pADR (R&D systems, 4335-MC-100, 1:1000 dilution).

**Drug treatment and X-ray radiation.** Cells were treated with 10 μM olaparib 24 h before or after 1 nM rapamycin treatment. Bafilomycin A1 at 300 nM for 6 h was used to inhibit autophagy. DNA damage was induced by 8 Gy gamma-irradiation.

**Clonogenic assays.** Cells were seeded at 500 cells/well for C4-2B and PC-3 and 1000 cells/well for LNCaP in 6-well plates and allowed to adhere for 24 h in 5% $CO_2$ at 37 °C. Medium was removed and replaced with RPMI complete medium containing olaparib (0.05 μM–20 μM). After 7 days of treatment, cells were fixed with methanol and stained with a solution of 50% v/v methanol and 0.5% m/v blue methylene (Sigma-Aldrich Inc.). Colonies were counted under a stereomicroscope and reported as a percentage of the control. $IC_{50}$ values were determined by using GraphPad Prism 8 software (GraphPad Software Inc., San Diego, CA). Each experiment was performed in duplicate and repeated three times.

**Protein preparation and western blot analysis**. Proteins were extracted from cell lines using mammalian protein extraction reagent (MPER; 50 nM Tris-HCl, 200 mM NaCl, 0.25% Triton 100X, and 10% glycerol) containing a protease and phosphatase inhibitor cocktail (Thermo Fisher Scientific, PIA32961). Protein concentration was determined by Bradford assay (Bio-Rad Laboratories, 500-0006). Twenty micrograms of total protein extract were separated in precast 4–15% gradient Tris-glycine SDS-polyacrylamide gels (Mini-PROTEAN® TGX™ Precast Gels, Bio-Rad, 456-1086) and transferred onto Trans-Blot Turbo Mini 0.2 μm nitrocellulose membranes (Bio-Rad, 170-4159). Membranes were blocked with 5% milk in PBS-Tween for 1 h and probed with primary antibodies overnight at 4 °C with agitation. Primary antibodies were detected with peroxidase-conjugated secondary antibodies Goat anti-mouse (Millipore, AP124P, 1:4000 dilution), Goat anti-rabbit (Millipore, AP156P, 1:10,000 dilution), and Rabbit anti-goat (Millipore, AP106P, 1/4000 dilution) and enhanced with chemiluminescence (Millipore, RPN2232) detected using the ChemiDoc MP Imaging System (Bio-Rad). Actin was used as a loading protein control. Each experiment was repeated three times. Image J was used to quantify western blot.

**Measurement of autophagic flux**. Cells were seeded onto coverslips at 15,000 cells/well for C4-2B and PC-3 and 25,000 cells/well for LNCaP in 24-well plates. After 24 h, cells were transfected with the Premo Autophagy Tandem Sensor RFP-GFP-LC3B BacMam 2.0 Expression vector (Thermo Fisher, P36239) according to manufacturer's protocol and were then used for experiments as described. Cells were fixed with formalin for 15 min at room temperature, washed using PBS, and coverslips were mounted onto slides using Prolong Gold® anti-fade reagent with DAPI (Life Technologies Inc., 14209 S). Three different confocal images per condition were obtained on a Leica TCS-SP5 inverted microscope using a HCX PL APO CS 63x/1.4 Oil UV objective. Excitation was performed using a 405 diode laser for DAPI, a 488 nm line of an Argon laser for GFP, and a 561 nm DPSS laser for RFP using a sequential acquisition at 400 Hz scan speed. Detection bandwidth was 415–478 nm for DAPI using a PMT, 498–551 nm for GFP using a HyD under the Standard mode and 5561–677 nm for RFP using a HyD under the Standard mode. Images were acquired with the Las-AF software. Final images are 8bits, 2048 × 2048 (axial pixel size of 120 nm). Z-stacks were performed to generate a maximum intensity projection (MIP) for a representative sampling of the thickness of each cell (10 z sections). Images were analyzed using FIJI software (NIH) with a macro, adapted from Daniel J. Shiwarski (creator, B.S., University of Pittsburgh). A mean of 20 cells per condition was quantified. Each condition was performed in triplicate and repeated three times.

**Cell cycle analysis**. Cells were seeded in 6-well plates and treated with rapamycin 24 h prior to olaparib treatment. Cells were fixed in 70% ethanol and incubated with 100 μg/mL RNase A and 25 μg/mL propidium iodide (PI). A maximum of 30,000 events was counted per condition using the Fortessa flow cytometer (BD Biosciences) and analyzed with FlowJo software.

**Immunocytochemistry**. A total of 8000–20,000 cells were seeded onto coverslips in 12-well plates and grown for different time points (8 and 16 h and 1, 2, and 6 days). Cells were fixed with formalin for 15 min at RT. Slides were blocked for 1 h at room temperature in PBS containing 1% BSA, 4% donkey serum, for SQSTM1/p62 and H2AX proteins, and 1% BSA, 20% FBS for Rad51/BRCA1. Coverslips were incubated with primary antibodies diluted in the same blocking buffer overnight at 4 °C. Cells were washed with PBS and incubated with appropriate secondary antibodies for 1 h at RT, anti-mouse Cy5 (Life Technologies Inc., A10524, 1:800 dilution), and anti-rabbit 488 (Thermo Fisher Scientific, A-11008, 1:500 dilution). Coverslips were mounted onto slides using Prolong Gold® anti-fade reagent with DAPI (Life Technologies Inc., P36935). For Rad51-BRCA1 and H2AX proteins, images (40X and 20X magnification, respectively) were obtained using a Zeiss microscope (Carl Zeiss, Zeiss AxioObserver Z1,). AxioVision™ software (Carl Zeiss) was used to calculate the average number of foci per nucleus. For SQSTM1/p62 protein, confocal images were acquired on a Leica TCS-SP5 inverted microscope using a HCX PL APO CS 63x/1.4 Oil UV objective. Excitation was performed using a 405 diode laser for DAPI, and a 633 nm HeNe laser using a sequential acquisition at 400 Hz scan speed. Detection bandwidth was 415–478 nm for DAPI using a PMT and 643–750 nm using a HyD under the Standard mode. Images were acquired with the Las-AF software. The final images are 8 bits, 1024 × 1024 with a zoom factor 2 (axial pixel size of 120 nm). Z-stacks were performed to generate a maximum intensity projection (MIP) for a representative sampling of the thickness of each cell (6–8 z sections). Images were analyzed using FIJI software (NIH) and Imaris software (Oxford Instruments).

**IncuCyte phase-contrast live-cell imaging assay**. Cells were seeded at 1000 cells/well for C4-2B and PC-3 and 1500 cells/well for LNCaP in 96-well plates. Cells were incubated with 1 nM rapamycin for 24 h before or after the start of a 6 day 10 μM olaparib treatment. Cell numbers to monitor proliferation were imaged by phase-contrast using the IncuCyte™ Live-Cell Imaging System (IncuCyte HD) at 2 h intervals from two separate regions. Each experiment was performed in triplicate and repeated three times.

**Analysis of HR and NHEJ activity**. HR and NHEJ activities were measured using the reporter plasmids pcDNA-GFP HR and pCDNA-GFP NHEJ, a gift from Dr. Jean-Yves Masson (Université de Laval, QC, Canada). One million cells of C4-2B, PC-3, and LNCaP were transfected using Lipofectamine 3000 (Thermo Fisher Scientific) with the reporter plasmid and pCMV-I-SceI vector containing an mCherry-tag. After 48 h, cells were collected, and the number of GFP-positive and mCherry-positive cells was determined by Fortessa flow cytometer (BD Biosciences). Quantification of GFP-positive cells (Q2) and GFP-positive + mCherry-positive cells (Q3) was achieved using Flowjo software. HR and NEHJ efficiency were determined following this equation: (1) $\frac{Q2}{Q2+Q3}$.

**Isolation of nuclear and cytoplasmic extract**. Nuclear extraction was prepared using NE-PER Nuclear and Cytoplasmic Extraction Kit (Thermo Fisher Scientific, 78833) according to the manufacturer's instructions. Briefly, treated cells were washed to PBS 1X and centrifuged at $1000 \times g$ for 5 min. The cell pellet was suspended into 35 μL of cytoplasmic reagent I by vortexing and incubated on ice. After 10 min incubation 1.9 μL of cytoplasmic reagent II was added, tubes were vortexed and incubated for 1 min on ice. The cytoplasmic extract was isolated by 5 min centrifugation at $16,000 \times g$ and transferred to a pre-chilled tube. To remove all cytoplasmic content, the insoluble pellet fraction was washed three times with ice-cold PBS 1X and centrifuged as previously. This pellet was next resuspended in 18 μL of nuclear extraction reagent by vortexing 20 s every 10 min, for a total of 40 min. Tubes were stocked at −80 °C until their centrifugation at $16,000 \times g$ for 10 min before western blotting. Cytoplasmic and nuclear fractions for each cell line were loaded in the same gel to ensure the purity of the nuclear fraction. Image J was used for quantification.

**Statistics and reproducibility**. Statistical analyses were performed using Graph Pad Prism 8 by the two-tail Student *t*-test, which was justified appropriately for every experimental design. The data were normally distributed and the variance between groups that were statistically compared was similar. A *p*-value of <0.05 was considered statistically significant. For all data, the mean ± SEM of three independent experiments is shown.

**Reporting summary**. Further information on research design is available in the Nature Research Reporting Summary linked to this article.

## Data availability

Source data presented in the figures is provided as a Source Data file. Additional relevant source data supporting this study is available from the corresponding authors upon request. Supplementary figures are available as a Supplementary Information file.

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

## Acknowledgements

We acknowledge F.S. and A.M.M.M. laboratory members for comments and discussion. We thank Dr. Gleave (Vancouver Prostate Centre) for C4-2B cell line, Dr. Russel (University of Ottawa) for CRISPR guide RNA against Atg16L1 and for Atg16L1-HA plasmid and. Dr. Yves Masson (Centre de recherche du CHU de Québec-Université de Laval) for pCMV-I-SceI, pcDNA-GFP HR/NHEJ. We also thank Dr. Cleret-Buhot from CRCHUM live imaging platform for her expertize, Jacqueline Chung for manuscript editing and the Institut du cancer de Montréal (ICM). This work was supported by Raymond Garneau Chair in Prostate Cancer Research of the Université de Montréal (F.S.), research grants from the Canadian Urological Association and Canderel/ICM and. Département de biologie moléculaire de l'Université de Montréal.

## Author contributions

M.C., B.P., H.F., A.M.M.M., and F.S. conceived and designed the experiments. M.C. and P.L. conducted the experiments. B.P., A.M.M.M., and F.S. supervised the project. M.C., B.P., H.F., A.M.M.M., and F.S. wrote the paper.

## Competing interests

F.S. is an advisory board member (personal) and has received research funding (institutional) from AstraZeneca, Pfizer, Astellas, Bayer and Janssen.
