## [Peer Review File · Communications Biology]

Reviewers' comments:

Reviewer #1 (Remarks to the Author):

Comments to the authors:

The Authors demonstrated that autophagy plays a role in the cellular response to olaparib treatment in pancreatic cancer cell lines. Such role might be mediated by the regulation of DNA repair efficiency, in particular through the inhibition of SQSTM1/p62 nuclear localization and the subsequent increase of FLNA, which is directly involved in the recruitment of proteins belonging to the HR pathway. The manuscript is well written and provides potential translational applications, such as the evaluation of autophagy basal levels as a strategy to predict patient responsiveness to PARPi therapy and the introduction of combination therapies with PARPi and autophagy inhibitors to overcome PARPi resistance.

However, the following concerns should be addressed:

- The results reported in Figure 3 showed a consistent increase of the percentage of cells in S phase upon Olaparib treatment. This does not seem in line with previous literature, showing the effect of this PARP inhibitor as an increase of G2/M phase cells at the expense of S phase (see Cheng Wu et al. Mol Cancer Ther 2021; Andreidesz K et al. Int J Mol Sci. 2021; Vescarelli E. et al. J Exp Clin Cancer Res. 2020; Camero S et al. J Cancer Res Clin Oncol. 2019). Such discrepancies should be addressed in the discussion.
- Moreover, the FACS analysis of cell cycle distribution reported a very low percentage of cells in S phase in the control, which is unusual for cancer cell lines. Please clarify these findings.
- Probably, to confirm the effect of Olaparib on cell cycle distribution, it would be useful to check by WB experiments the expression of proteins related to cell cycle checkpoints, such as Cyclin B1, Cdc25C, p21 and Cyclin D1
- The authors should add more details in the discussion, focusing on current therapies for PC and the impact of PARPi resistance on patient survival. This would help the reader to understand the significance of findings illustrated in this study

Reviewer #2 (Remarks to the Author):

In this manuscript, the authors use LNCaP, C4-2B, and PC-3 prostate cancer cells lines to determine how autophagy causes resistance to olaparib treatment. They use rapamycin to induce autophagy, and CRISPR KO of ATGp16L1 to block autophagy. Based on their results using these two manipulations, they propose a model whereby pre-activation of autophagy by rapamycin increases the sensitivity to olaparib by decreasing nuclear SQSTM1/p62, which increases homologous recombination-mediated repair through increased filamin A expression and recruitment of BRCA1/Rad51.

The data outlined in Figure 1 nicely demonstrates a strong correlation between high autophagy levels and increased resistance to olaparib in these three cell lines, and that inhibiting autophagy sensitizes to olaparib. However, the concept that autophagy mediates resistance to olaparib is not necessarily novel and the proposed pathways have been previously demonstrated in other cancer models. The data is strictly correlative with respect to Rad51/Brca2, p62, and filamin A and their presumed dependency/relationship is based on associations previously reported in the literature. The lack of any in vivo data to support a treatment strategy also weakens the potential significance.

The majority of the manuscript is focused on comparing what happens when these cell lines are pretreated with rapamycin 1 day before olaparib vs treated 1 day after olaparib. The strongest data using this approach is demonstrating that rapamycin pretreatment causes a significant increase in cell proliferation, whereas delayed rapamycin pretreatment does not do this and KO of ATG16L1 abrogates this proliferative response, although to a lesser degree in PC3 – consistent with its increased resistance. The other highly supportive data is using an HR vs NHEJ reporter assay and finding that pre-treatment with rapamycin leads to increased HR, but not NHEJ activity. This HR response was blocked by loss of ATG16L1, thus linking autophagy specifically to the HR

response.

There are several major concerns about the rest of the data. Unfortunately, the rest of the data is not consistent between the cell lines nor with repeat experiments shown in supplementary data. The authors claim that in Fig 2B and S1B, rapamycin pretreatment, but not posttreatment leads to increased LC3II and decreased p62 in all lines. This is seen in LNCaP and C4-2B in Fig 2B, but only for LNCaP in Fig S1B and never for PC3. Furthermore, rapamycin or olaparib alone is often having the same impact as the combined treatment. Same applies to the cell cycle analysis (Fig. 2D, S1C), where C4-2B seems to act as predicted, but the other lines are inconsistent.

Data in figures 3 and 5 don't include the post-treatment data to be able to directly compare to the pre-treatment data within the same experiment. Instead the post-treatment data is placed with another set of data in supplementary figures 2,3, and 4. The quantification between these different sets of data leads to differences in results and differences in statistical significance. Interpretation was also complicated by the fact that Supplementary Fig 3 data seems to be superimposed over Supplementary Fig 2 data. This questions whether the stats presented are simply technical replicates within each experiment and not biological replicates.

Data in Figure 5C would be strengthened by demonstrating that p62 is also nuclear localized in these extracts.

It is not clear why the quantification of the blots in Fig5C were normalized to olaparib rather than to untreated controls. The effect on filamin A is levels is modest at best. Quantification of several different blots from different experiments is required to assure statistical significance and consistency.

Overall these results are only correlative with respect to Rad51/Brca2, p62, and filamin A and their presumed dependency/relationship. There are no experiments that disrupt expression/activity of any of these proteins and show that they impact the response to olaparib. Authors need to be careful about using strong statements indicating their data support their proposed model.

The authors should also acknowledge that blocking ATG16L1 or autophagy in general will result in p62 accumulation simply because you are blocking its degradation.

It should be noted that C4-2B and LNCaP display a dramatic response to olaparib (log-fold) upon autophagy inhibition, but PC-3 displays less than a 2-fold increase in sensitivity, indicating at least for PC-3 cells, there are other factors involved. The authors need to acknowledge and address this more in the Discussion.

Reviewer #3 (Remarks to the Author):

Cahuzac et al investigated the impact of autophagy on prostate cancer cell response to the PARPi olaparib by following the autophagy activation timeline. And Cahuzac et al define that pre-activation of autophagy before olaparib treatment reduces the level of SQSTM1/p62 in the nucleus, results in an increase of DNA repair activity by homologous recombination to repair double-strand breaks induced by olaparib, and enhances cell proliferation. The authors should design and perform more experiments to support their conclusions.

1. The authors showed that PC-3 with a higher basal level of autophagy is resistant to olaparib compared to LNCaP and C4-2B with a lower basal level of autophagy (Fig. 1A-C). How about the basal expression level of PARP1/2, which may also affect the antiproliferation effect of olaparib?
2. The authors showed that AR-negative PC cells may have a higher basal level of autophagy and are olaparib-resistant compared to AR-positive PC cells (Fig. 1A-C). But the authors only chose one AR-negative PC cell line PC-3, and also didn't include 22Rv1 cell line that is AR-V7-positive. How about the basal level of autophagy and the IC50 values for olaparib in PC-3 and 22Rv1 cells?
3. The authors mentioned "Both RO10 and O10R conditions showed similar levels of LC3-II,

indicating similar levels of autophagy activation" (Fig. 2B). The cell proliferation was significantly increased under RO10 conditions, but not under O10R conditions (Fig. 2C). Even under O10R conditions, the proliferation of LNCaP, C4-2B and PC-3 cells was decreased compared to olaparib treatment alone (Fig. 2C). The authors should perform another experiment (may choose cell viability assay using CellTiter-Glo) to further confirm the different anti-proliferation effect under RO10 and O10R conditions, which both showed a similarly high level of autophagy activation.

4. In Fig.2B, the western blots showed that in C4-2B WT cells, the autophagy was not activated, even was inhibited in all the groups with Rapamycin treatment, compared to the Control group as shown by LC3-II and p62. The authors should further confirm that. For these western blots, the authors may also show the normalized values or plots explaining the change in expression normalized to Actin.

5. In Fig. 2C, it showed that the olaparib treatment (O10) inhibited the proliferation of PC-3 Atg16L1 KO cells more than that of PC-3 WT cells. But in Fig. 1G-H, the authors have demonstrated that complete depletion of autophagy reduced the olaparib IC50 value from 2.13 μ M to 1.5 μ M for PC-3. The authors should further confirm that.

6. The authors showed that pre-activation of autophagy limited the effect of olaparib on cell proliferation and cell cycle and this protective effect was abrogated when autophagy was abrogated by Atg16L1 KO (Fig.2). The authors should do one more knockout rescue experiment to see if the re-activation of autophagy would make the PC cells sensitive to olaparib again.

7. The authors showed that using their plasmid-based DNA repair reporter assays, complete depletion of autophagy by Atg16L1 KO significantly reduced the efficiency of HR (Fig.4F). Could the efficiency of HR be rescued with re-expression of Atg16L1?

8. The authors mentioned "we observed that the KO cell lines had higher expression levels of SQSTM1/p62 compared to WT cells (Fig. 2B)" in line 275. because the western results of SQSTM1/p62 protein in WT and KO cell lines were shown on the different blots separately in Fig. 2B, it's not clear if the KO cell lines had higher expression levels of SQSTM1/p62. For comparison of the expression levels of SQSTM1/p62, the authors should show the samples on the same one blot, and may also show the normalized values or plots explaining the difference in expression normalized to Actin.

9. In Fig. 5C, it seems the loadings of samples are not equal as shown by loading control SP1 and tubulin. The authors should normalize the bands of FLNA to loading control SP1 or tubulin. And the authors should also include SQSTM1/p62 in Fig. 5C to visually show the change of FLNA along with p62 in the nucleus.

Minor points:

1. In Fig. 2B, there showed two bands for p62 in LNCaP WT and PC-3 WT cells, but one band for p62 in C4-2B WT, LNCaP KO Atg16L1, C4-2B KO Atg16L1, and PC-3 KO Atg16L1 cells.

2. In Fig. S2, there is no panel C and D. Both Fig. S2C and S2D are missed.

3. The whole Fig.S3 is missed.

Reviewer #1 (Remarks to the Author):

Comments to the authors:

The Authors demonstrated that autophagy plays a role in the cellular response to olaparib treatment in pancreatic cancer cell lines. Such role might be mediated by the regulation of DNA repair efficiency, in particular through the inhibition of SQSTM1/p62 nuclear localization and the subsequent increase of FLNA, which is directly involved in the recruitment of proteins belonging to the HR pathway. The manuscript is well written and provides potential translational applications, such as the evaluation of autophagy basal levels as a strategy to predict patient responsiveness to PARPi therapy and the introduction of combination therapies with PARPi and autophagy inhibitors to overcome PARPi resistance.

However, the following concerns should be addressed:

- 1. The results reported in Figure 3 showed a consistent increase of the percentage of cells in S phase upon Olaparib treatment. This does not seem in line with previous literature, showing the effect of this PARP inhibitor as an increase of G2/M phase cells at the expense of S phase (see Cheng Wu et al. Mol Cancer Ther 2021; Andreidesz K et al. Int J Mol Sci. 2021; Vescarelli E. et al. J Exp Clin Cancer Res. 2020; Camero S et al. J Cancer Res Clin Oncol. 2019). Such discrepancies should be addressed in the discussion.**

While we agree that the effect of olaparib has been associated in some studies with an increase in cells within the G2/M phase of the cell cycle, there is also a literature that clearly identifies conditions where the S phase is favored. Mani and al. showed that 25 uM of olaparib induced a blockade in G2/M in MDAMB231 but a blockade in S phase for MDAMB453.

To address this point, we have modified the manuscript to include the appropriate reference as follows in line 169: *In C4-2B cells, after 6 days of O10 treatment, 70% of cells were blocked in S phase compared to 7% of control cells as it was shown in different cell lines in the literature (Yang, Ndawula et al. 2015, Pirotte, Holzhauser et al. 2018, Mani, Jonnalagadda et al. 2019).*

- 2. Moreover, the FACS analysis of cell cycle distribution reported a very low percentage of cells in S phase in the control, which is unusual for cancer cell lines. Please clarify these findings.**

While there are instances where the distribution in S phase is high, it is not without precedence to see the proportion of cancer cells in S phase vary between 5-30% depending on the cell lines used and cellular confluence, especially in time course experiments (Bort, Quesada et al. 2018, Sun, Weng et al. 2018, Wu, Peng et al. 2021). Therefore, the percentages we observed (15% for LNCaP, 7% for C4-2B and 10% PC-3 at day 2). Since autophagy is compared between the experimental groups and not the untreated control, we feel confident in our conclusions.

- 3. Probably, to confirm the effect of Olaparib on cell cycle distribution, it would be useful to check by WB experiments the expression of proteins related to cell cycle checkpoints, such as Cyclin B1, Cdc25C, p21 and Cyclin D1**

While we agree that the suggestion is interesting in itself, it would not impact the conclusions of the study. In addition, the relationship between these proteins and the cell cycle have been well documented and thus no additional experiments were performed.

- 4. The authors should add more details in the discussion, focusing on current therapies for PC and the impact of PARPi resistance on patient survival. This would help the reader to understand the significance of findings illustrated in this study.**

This would indeed help the reader better understand the impact of our findings. We determined that the information could be introduced early on to better contextualize our work. Therefore, we added the following paragraph in the introduction to clarify this point between line 45 and 56.

Early-stage prostate cancer has an excellent prognosis with excellent 5- and 10-year overall survival with local therapy +/- androgen deprivation therapy (ADT). In advanced prostate cancer, especially when patients become resistant to ADT (known as castration resistant prostate cancer or CRPC), the available therapeutic options are non-curative and survival is generally less than 3 years. Therapeutic options include taxane-based chemotherapy and more recently novel hormonal therapies, that directly or indirectly target the androgen receptor, such as abiraterone and enzalutamide. Eventually patients develop resistance to available therapeutic options and succumb to their disease. Ongoing research continues to better understand and develop therapeutic approaches in patients who fail novel hormone therapies. One avenue of intense research in this area is in the use of PARP inhibitors in prostate cancer.

Reviewer #2 (Remarks to the Author):

In this manuscript, the authors use LNCaP, C4-2B, and PC-3 prostate cancer cells lines to determine how autophagy causes resistance to olaparib treatment. They use rapamycin to induce autophagy, and CRISPR KO of ATGp16L1 to block autophagy. Based on their results using these two manipulations, they propose a model whereby pre-activation of autophagy by rapamycin increases the sensitivity to olaparib by decreasing nuclear SQSTM1/p62, which increases homologous recombination-mediated repair through increased filamin A expression and recruitment of BRCA1/Rad51.

The data outlined in Figure 1 nicely demonstrates a strong correlation between high autophagy levels and increased resistance to olaparib in these three cell lines, and that inhibiting autophagy sensitizes to olaparib. However, the concept that autophagy mediates resistance to olaparib is not necessarily novel and the proposed pathways have been previously demonstrated in other cancer models. The data is strictly correlative with respect to Rad51/Brca2, p62, and filamin A and their presumed dependency/relationship is based on associations previously reported in the literature. The lack of any *in vivo* data to support a treatment strategy also weakens the potential significance.

The majority of the manuscript is focused on comparing what happens when these cell lines are pretreated with rapamycin 1 day before olaparib vs treated 1 day after olaparib. The strongest data using this approach is demonstrating that rapamycin pretreatment causes a significant increase in cell proliferation, whereas delayed rapamycin pretreatment does not do this and KO of ATG16L1 abrogates this proliferative response, although to a lesser degree in PC3 – consistent with its increased resistance. The other highly supportive data is using an HR vs NHEJ reporter assay and finding that pre-treatment with rapamycin leads to increased HR, but not NHEJ activity. This HR response was blocked by loss of ATG16L1, thus linking autophagy specifically to the HR response.

- 1. There are several major concerns about the rest of the data. Unfortunately, the rest of the data is not consistent between the cell lines nor with repeat experiments shown in supplementary data. Same applies to the cell cycle analysis (Fig. 2D, S1C), where C4-2B seems to act as predicted, but the other lines are inconsistent.**

While we agree that the three cell lines when compared to each other are distinct in the level of response, we would point out that the phenotype described is internally consistent within each of the cell lines, although the effect is less marked in the LNCaP and PC-3 cell lines. To make this point more clearly, we have modified the following sentence of the discussion in line 368:

We hypothesize that this different level of autophagy-mediated resistance must be due to different factors such as AR expression, basal level of autophagy, *or even the different degrees of autophagy activation by rapamycin.*

- 2. The authors claim that in Fig 2B and S1B, rapamycin pretreatment, but not posttreatment leads to increased LC3II and decreased p62 in all lines. This is seen in LNCaP and C4-2B in Fig 2B, but only for LNCaP in Fig S1B and never for PC3. Furthermore, rapamycin or olaparib alone is often having the same impact as the combined treatment.**

We thank the reviewer for this comment but we would like to note that we did not indicate that there is difference between LC3II and p62 between the pre and post treatment. Indeed, the

following sentence can be found in the result section: “Both RO10 and O10R conditions showed similar levels of LC3-II, indicating similar levels of autophagy activation for day 2 and 6”. For clarity we have modified the sentence as follows between line 151 and 153: *No significant differences were observed for LC3-II expression in RO10 and O10R conditions indicating similar levels of autophagy activation (Fig. 2b and S2b)*. In addition, as there seems to be confusion around this point, we have added the quantification of LC3-II expression in Fig. 2c and S2b to clearly show that autophagy in pre- or post- activated populations is the same. It supports the notion that the effect we observed is due to the timeline of treatment and not different level of autophagy between these two conditions. While we agree that olaparib alone can have an impact on autophagy, in this study we are focusing on the impact of induced autophagy on the response to olaparib and therefore cell fates post-olaparib would be the same in all conditions.

- 3. Data in figures 3 and 5 don't include the post-treatment data to be able to directly compare to the pre-treatment data within the same experiment. Instead the post-treatment data is placed with another set of data in supplementary figures 2,3, and 4. The quantification between these different sets of data leads to differences in results and differences in statistical significance. Interpretation was also complicated by the fact that Supplementary Fig 3 data seems to be superimposed over Supplementary Fig 2 data. This questions whether the stats presented are simply technical replicates within each experiment and not biological replicates.**

In Figure 2 we included all experimental conditions so that the reader could appreciate the effect of the RO10 treatment. As the phenomena was consistent in subsequent figures, we chose to present the O10R treatments as supplementary material in order to unencumber the subsequent figures. As it was unclear that each experiment was a biological replicate we have now included this information in the supplementary figure legend. We regret the error in superimposing Supplementary Fig 2 and 3, we have resolved this issue in the resubmission.

- 4. Data in Figure 5C would be strengthened by demonstrating that p62 is also nuclear localized in these extracts. It is not clear why the quantification of the blots in Fig5C were normalized to olaparib rather than to untreated controls. The effect on filamin A is levels is modest at best. Quantification of several different blots from different experiments is required to assure statistical significance and consistency.**

We thank the reviewer for this suggestion and agree that this information would strengthen our manuscript. We added a western blot tracking SQSTM1/p62 localization in the nucleus and the quantification from three independent experiments now are reported in Fig. 5C. We chose to compare RO10 and O10R to O10 alone as the main purpose of this experiment was to show how autophagy pre-activation/post-activation affects the nuclear level of FLNA and SQSTM1/p62 in the O10 condition. We added the following sentence in the results to address this comment between line 288 and 298:

RO10 and O10R conditions were compared with O10 as we studied the impact of autophagy pre- and post-activation on FLNA and SQSTM1/p62 nuclear expression after an olaparib treatment. Under autophagy pre-activation (RO10), FLNA was more expressed in LNCaP WT (2.3-fold change), C4-2B WT (3.7-fold change) and PC-3 WT (3.4-fold change) in the nucleus compared to O10 (Fig. 5c-d), which aligned with increased Rad51/BRCA1 foci. In contrast, O10R conditions or KO cell lines did not show this increase in FLNA expression under RO10 conditions. This increase of FLNA was accompanied by a significant decrease of SQSTM1/p62 in the nucleus in LNCaP WT (3.3-fold change), C4-2B WT (3.4-fold change) and PC-3 WT (4.5-fold change) in RO10 conditions compared to O10 (Fig. 5c and e). This effect was not observed in O10R condition and in PC KO Atg16L1 cell lines.

5. **Overall these results are only correlative with respect to Rad51/Brc2, p62, and filamin A and their presumed dependency/relationship. There are no experiments that disrupt expression/activity of any of these proteins and show that they impact the response to olaparib. Authors need to be careful about using strong statements indicating their data support their proposed model.**

We thank the reviewer for this comment and agree that it would be beneficial to provide direct evidence for this relationship. We conducted supplementary experiments where we modulated p62 expression using a siRNA approach. This demonstrated that pre-siRNA knock-down of SQSTM1/p62 (siO10) in the PC KO Atg16L1 cell lines had similar effects as RO10 on cell proliferation (Fig. 6c and S9a), on H2AX foci resolution (Fig. 6d and S9b) and HR efficiency (Fig. 6e). Post-transfection (O10si) had no effects as O10R. In addition to these new figures we added the following text in the results between line 303 and 329:

Targeting SQSTM1/p62 rescued effect of autophagy in PC KO Atg16L1 cell lines

To confirm the importance of SQSTM1/p62 in this autophagy-mediated resistance, we used a siRNA against SQSTM1/p62 in our PC KO Atg16L1 cell lines and in WT ones (Fig. 6 and S9). We followed the sequence of treatment as rapamycin, by transfecting siRNA (si) or scramble (Sble) 24 hours before or after olaparib treatment (Sble/siO10, O10Sble/si; respectively) (Fig. S9a). We confirmed siRNA efficacy by western blot and observed an important decrease of SQSTM1/sip62 protein level mainly in PC KO Atg16L1 but also in PC WT cell lines at day 2 and 6 (Fig. 6a). Interestingly, pre-inhibition of SQSTM1/p62 (siO10) in LNCaP, C4-2B and PC-3 KO reverses effects of autophagy depletion on cell proliferation after olaparib treatment (15% to 37%, 16% to 52% and 12% to 38%, respectively), where no significant differences was previously observed in RO10 conditions (Fig. 6b and 2d). A decrease of SQSTM1/p62 had a similar effect as autophagy pre-activation

by rapamycin (RO10), an increase of cell proliferation compared to O10 by 1.7 to 2-fold change for WT cell lines. This phenotype was lost when inhibition of SQSTM1/p62 was performed after olaparib treatment (O10si). No significant differences were also observed in SbleO10 and O10Sble conditions (Fig. S9b). Pre-inhibition of SQSTM1/p62 in KO Atg16L1 cell lines also decreased the number of γ -H2AX foci per cell after 2 days compared to cells in O10 treatment (LNCaP KO, 8 vs. 22, $p=0.008$; C4-2B KO, 10 vs. 21, $p=0.027$; and PC-3 KO, 11 vs. 18, $p=0.008$) (Fig. 6c). A similar decrease was observed in WT cells. In all Sble and O10si conditions, levels of γ -H2AX foci were similar from O10 conditions for all cell lines (Fig. S9c). To determine if this decrease in γ -H2AX foci was due to an increase of HR efficiency, we used our GFP reporter assay (Fig. 6d). As expected, HR was more efficient in KO Atg16L1 and WT PC cell lines where SQSTM1/p62 was pre-inhibited (siO10) and not O10si conditions compared to SbleO10 and O10Sble, respectively. This provides evidence that the regulation of SQSTM1/p62 drives the autophagy-mediated resistance observed when autophagy was pre-activated in PC WT cell lines.

- 6. The authors should also acknowledge that blocking ATG16L1 or autophagy in general will result in p62 accumulation simply because you are blocking its degradation.**

We agree and clarified this point as follow in line 270: "As expected, we observed that the KO cell lines had higher expression levels of SQSTM1/p62 compared to WT cells (Fig. 2B)."

- 7. It should be noted that C4-2B and LNCaP display a dramatic response to olaparib (log-fold) upon autophagy inhibition, but PC-3 displays less than a 2-fold increase in sensitivity, indicating at least for PC-3 cells, there are other factors involved. The authors need to acknowledge and address this more in the Discussion.**

Clarity around this point has already been addressed in our response to Reviewer 1.

Reviewer #3 (Remarks to the Author):

Cahuzac et al investigated the impact of autophagy on prostate cancer cell response to the PARPi olaparib by following the autophagy activation timeline. And Cahuzac et al define that pre-activation of autophagy before olaparib treatment reduces the level of SQSTM1/p62 in the nucleus, results in an increase of DNA repair activity by homologous recombination to repair double-strand breaks induced by olaparib, and enhances cell proliferation. The authors should design and perform more experiments to support their conclusions.

- 1. The authors showed that PC-3 with a higher basal level of autophagy is resistant to olaparib compared to LNCaP and C4-2B with a lower basal level of autophagy (Fig. 1A-C). How about the basal expression level of PARP1/2, which may also affect the antiproliferation effect of olaparib?**

We thank the reviewer for raising an interesting point. We addressed this by performing an additional western blot (Fig. S1) to determine the expression of PARP1 and PARylation. We observed that depletion of autophagy did not impact PARP1 expression and PARylation. The following sentence has been added to the results to address this point between line 140 and 143:

To ensure that PARP1 or PARylation did not affect olaparib sensitivity, we measured basal level protein by western blot and found no differences significant difference between WT PC-3 cells and those undergoing autophagy depletion (Fig. S1).

- 2. The authors showed that AR-negative PC cells may have a higher basal level of autophagy and are olaparib-resistant compared to AR-positive PC cells (Fig. 1A-C). But the authors only chose one AR-negative PC cell line PC-3, and also didn't include 22Rv1 cell line that is AR-V7-positive. How about the basal level of autophagy and the IC50 values for olaparib in PC-3 and 22Rv1 cells?**

We have unpublished results that measure basal level of autophagy (western blot) and olaparib sensitivity (clonogenic) in the 22Rv1 cell line. We observed similar levels of autophagy as LNCaP and C4-2B and 22Rv1 cells had an olaparib IC50 of 0.94 μ M. As AR-V7 expression could be a confounding variable, we decided not to continue with this cell line.

- 3. The authors mentioned "Both RO10 and O10R conditions showed similar levels of LC3-II, indicating similar levels of autophagy activation" (Fig. 2B). The cell proliferation was significantly increased under RO10 conditions, but not under O10R conditions (Fig. 2C). Even under O10R conditions, the proliferation of LNCaP, C4-2B and PC-3 cells was decreased compared to olaparib treatment alone (Fig. 2C). The authors should perform another experiment (may choose cell viability assay using CellTiter-Glo) to further confirm the different anti-proliferation effect under RO10 and O10R conditions, which both showed a similarly high level of autophagy activation.**

We again thank the reviewer for this comment but we feel that our choice of proliferation assay is more judicious based on the phenomena we are trying to characterize. More specifically, we know that autophagy impact metabolism (production of ATP, mitochondrial activity (Guo, Teng

et al. 2016, Ferro, Servais et al. 2020)), and therefore the Incucyte system, which directly visualizes and quantifies proliferation, is more appropriate.

- 4. In Fig.2B, the western blots showed that in C4-2B WT cells, the autophagy was not activated, even was inhibited in all the groups with Rapamycin treatment, compared to the Control group as shown by LC3-II and p62. The authors should further confirm that. For these western blots, the authors may also show the normalized values or plots explaining the change in expression normalized to Actin.**

We agree and have added the quantification of LC3-II in Fig.2c and S2b to clarify the visualization of data.

- 5. In Fig. 2C, it showed that the olaparib treatment (O10) inhibited the proliferation of PC-3 Atg16L1 KO cells more than that of PC-3 WT cells. But in Fig. 1G-H, the authors have demonstrated that complete depletion of autophagy reduced the olaparib IC50 value from 2.13 μ M to 1.5 μ M for PC-3. The authors should further confirm that.**

While the observation the reviewer makes is correct, we would point out that the IC50 results are based on clonogenic assays which are difficult to directly compare to proliferation assays. However, the critical point is that the overall trend is consistent. In both cell proliferation and IC50 experiments we observed a decrease of these two parameters when autophagy was abrogated.

- 6. The authors showed that pre-activation of autophagy limited the effect of olaparib on cell proliferation and cell cycle and this protective effect was abrogated when autophagy was abrogated by Atg16L1 KO (Fig.2). The authors should do one more knockout rescue experiment to see if the re-activation of autophagy would make the PC cells sensitive to olaparib again.**

We agree that this result would strengthen the manuscript. We therefore confirmed our results by a rescue experiment as suggested using Atg16L1-HA and we reperformed cell proliferation assays. These results are now presented in Fig. S3 where we demonstrate that the rescue of Atg16L1 expression restore dynamics of autophagy and where RO10 conditions had the same effect on cell proliferation in this PC KO-rescue Atg16L1 as PC WT. We also did the experiment in WT-rescue Atg16L1 to ensure that rescue did not affect our previous observations. We also added the following text in the results between line 159 and 166:

We also rescued the depletion of autophagy by introducing a plasmid coding for Atg16L1 with a HA-tag in our PC WT and KO Atg16L1 cell lines (Fig. S3). Expression of Atg16L1-HA restored autophagy dynamics by the lipidation of LC3-I in LC3-II that was not observed in PC KO cell lines in autophagy induction and inhibition conditions (Fig. S3a and Fig. 1f). Expression of Atg16L1-HA had no effects on autophagy in WT cell lines. PC KO-rescue Atg16L1 also harbored a higher cell proliferation when autophagy was pre-activated (RO10) compared to PC

KO Atg16L1 (LNCaP, 48% vs. 23%; C4-2B, 39% vs. 19%; and PC-3, 72% vs. 24%) (Fig. S3b and Fig. 2d).

7. **The authors showed that using their plasmid-based DNA repair reporter assays, complete depletion of autophagy by Atg16L1 KO significantly reduced the efficiency of HR (Fig.4F). Could the efficiency of HR be rescued with re-expression of Atg16L1?**

We confirmed our results by a rescue experiment as suggested using Atg16L1-HA and we reperformed the measurement of DNA repair efficiency. These results can be found in Fig. S7 where we compared HR efficiency in PC KO-rescue Atg16L1 and PC WT compared to PC KO Atg16L1. Rescue of Atg16L1 permits PC KO-rescue Atg16L1 to restore HR efficiency to similar levels found in PC WT cells. The following text has been added in the results to address this point between line 259 and 261:

Rescue of autophagy using an Atg16L1-HA plasmid significantly restored the loss of HR efficiency in PC KO Atg16L1 cell lines (LNCaP KO-rescue, $p=0.0036$; C4-2B KO-rescue, $p=0.02$; and PC-3 KO-rescue, $p=0.0007$) (Fig. S7).

8. **The authors mentioned "we observed that the KO cell lines had higher expression levels of SQSTM1/p62 compared to WT cells (Fig. 2B)" in line 275. because the western results of SQSTM1/p62 protein in WT and KO cell lines were shown on the different blots separately in Fig. 2B, it's not clear if the KO cell lines had higher expression levels of SQSTM1/p62. For comparison of the expression levels of SQSTM1/p62, the authors should show the samples on the same one blot, and may also show the normalized values or plots explaining the difference in expression normalized to Actin.**

Following the reviewer suggestion, new western blots were performed where PC WT and PC KO were loaded on the same gel (new Fig. 2b). As the visualization of difference is clear we felt we did not have to add additional plots to clearly demonstrate the trend.

9. **In Fig. 5C, it seems the loadings of samples are not equal as shown by loading control SP1 and tubulin. The authors should normalize the bands of FLNA to loading control SP1 or tubulin. And the authors should also include SQSTM1/p62 in Fig. 5C to visually show the change of FLNA along with p62 in the nucleus.**

We have taken into account this comment. SP1 was used as a control of the nuclear fraction and tubulin as a control for the cytoplasmic fraction. We added SQSTM1/p62 expression in the nucleus in Fig. 5c and the quantification of FLNA and SQSTM1/p62 in Fig. 5d-e. As the nuclear fraction is the most critical, we have moved the cytoplasmic fraction to Fig. S8c and kept SP1, tubuline and Atg16L1 as controls. The following text has been added to the results:

We determined whether FLNA was important in this autophagy-mediated resistance to olaparib by measuring the nuclear fraction of FLNA and SQSTM1/p62 (Fig. 5c-d and S8c). RO10 and O10R conditions were compared with O10 as we studied the impact of autophagy pre- and

post-activation on FLNA and SQSTM1/p62 nuclear localisation after an olaparib treatment. Under autophagy pre-activation (RO10), FLNA was more highly localized to the nucleus in LNCaP WT (2.3-fold change), C4-2B WT (3.7-fold change) and PC-3 WT (3.4-fold change) compared to O10 (Fig. 5c-d), which correlates well with increased Rad51/BRCA1 foci. In contrast, O10R conditions or KO cell lines did not show this increase in FLNA expression under RO10 conditions. This increase of FLNA was accompanied by a significant decrease of SQSTM1/p62 in the nucleus in LNCaP WT (3.3-fold change), C4-2B WT (3.4-fold change) and PC-3 WT (4.5-fold change) in RO10 conditions compared to O10 (Fig. 5c and e). This effect was not observed in O10R condition and in PC KO Atg16L1 cell lines.

Minor points:

- a. In Fig. 2B, there showed two bands for p62 in LNCaP WT and PC-3 WT cells, but one band for p62 in C4-2B WT, LNCaP KO Atg16L1, C4-2B KO Atg16L1, and PC-3 KO Atg16L1 cells.
We are sorry for the confusion, stripping of Atg16L1 blot was not sufficient to remove all Atg16L1 in these two cell lines. The problem was solved in a new western blot.
- b. In Fig. S2, there is no panel C and D. Both Fig. S2C and S2D are missed. The whole Fig.S3 is missed.
We are sorry about the superimposition between Supplementary Fig 2 and 3, we have resolved this issue in the resubmission.

REFERENCES

Bort, A., et al. (2018). "Identification of a novel 2-oxindole fluorinated derivative as in vivo antitumor agent for prostate cancer acting via AMPK activation." *Sci Rep* **8**(1): 4370.

The key metabolic sensor adenosine monophosphate-dependent kinase (AMPK) has emerged as a promising therapeutic target for cancer prevention and treatment. Besides its role in energy homeostasis, AMPK blocks cell cycle, regulates autophagy and suppresses the anabolic processes required for rapid cell growth. AMPK is especially relevant in prostate cancer in which activation of lipogenic pathways correlate with tumor progression and aggressiveness. This study reports the discovery of a new series of 2-oxindole derivatives whose AMPK modulatory ability, as well as the antitumoral profile in prostate cancer cells, was evaluated. One of the assayed compounds, compound 8c, notably activated AMPK in cultured PC-3, DU145 and LNCaP prostate cancer cells. Likewise, compound 8c caused PC-3, DU145 and LNCaP cells viability inhibition. Selective knocking down of alpha1 or alpha2 isoforms as well as in vitro assays using human recombinant alpha1beta1gamma1 or alpha2beta1gamma1 AMPK isoforms revealed that compound 8c exhibit preference for AMPKalpha1. Consistent with efficacy at the cellular level, compound 8c was potent in suppressing the growth of PC-3 xenograft tumors. In conclusion, our results show that a new 2-oxindole fluorinated derivative exerts potent in vivo antitumor actions against prostate cancer cells, indicating a promising clinical therapeutic strategy for the treatment of androgen-independent prostate cancer.

Ferro, F., et al. (2020). "Autophagy and mitophagy in cancer metabolic remodelling." *Semin Cell Dev Biol* **98**: 129-138.

Metabolic reprogramming in tumours is now recognized as a hallmark of cancer, participating both in tumour growth and cancer progression. Cancer cells develop global metabolic adaptations allowing them to survive in the low oxygen and nutrient tumour microenvironment. Among these metabolic adaptations, cancer cells use glycolysis but also mitochondrial oxidations to produce ATP and building blocks needed for their high proliferation rate. Another particular adaptation of cancer cell metabolism is the use of autophagy and specific forms of autophagy like mitophagy to recycle intracellular components in condition of metabolic stress or during anticancer treatments. The plasticity of cancer cell metabolism is a major limitation of anticancer treatments and could participate to therapy resistances. The aim of this review is to report recent advances in the understanding of the relationship between tumour metabolism and autophagy/mitophagy in order to propose new therapeutic strategies.

Guo, J. Y., et al. (2016). "Autophagy provides metabolic substrates to maintain energy charge and nucleotide pools in Ras-driven lung cancer cells." *Genes Dev* **30**(15): 1704-1717.

Autophagy degrades and is thought to recycle proteins, other macromolecules, and organelles. In genetically engineered mouse models (GEMMs) for Kras-driven lung cancer, autophagy prevents the accumulation of defective mitochondria and promotes malignancy. Autophagy-deficient tumor-derived cell lines are respiration-impaired and starvation-sensitive. However, to what extent their sensitivity to starvation arises from defective mitochondria or an impaired supply of metabolic substrates remains unclear. Here, we sequenced the mitochondrial genomes of wild-type or autophagy-deficient (Atg7^{-/-}) Kras-driven lung tumors. Although Atg7 deletion resulted in increased mitochondrial mutations, there were too few nonsynonymous mutations to cause generalized mitochondrial dysfunction. In contrast, pulse-chase studies with isotope-labeled nutrients revealed impaired mitochondrial substrate supply during starvation of the autophagy-deficient cells. This was associated with increased reactive oxygen species (ROS), lower energy charge, and a dramatic drop in total nucleotide pools. While starvation survival of the autophagy-deficient cells was not rescued by the general antioxidant N-acetyl-cysteine, it was fully rescued by glutamine or glutamate (both amino acids that feed the TCA cycle and nucleotide synthesis) or nucleosides. Thus, maintenance of nucleotide pools is a critical

challenge for starving Kras-driven tumor cells. By providing bioenergetic and biosynthetic substrates, autophagy supports nucleotide pools and thereby starvation survival.

Mani, C., et al. (2019). "Prexasertib treatment induces homologous recombination deficiency and synergizes with olaparib in triple-negative breast cancer cells." *Breast Cancer Res* **21**(1): 104.

BACKGROUND: Breast cancer remains as one of the most lethal types of cancer in women. Among various subtypes, triple-negative breast cancer (TNBC) is the most aggressive and hard to treat type of breast cancer. Mechanistically, increased DNA repair and cell cycle checkpoint activation remain as the foremost reasons behind TNBC tumor resistance to chemotherapy and disease recurrence. **METHODS:** We evaluated the mechanism of prexasertib-induced regulation of homologous recombination (HR) proteins using 20S proteasome inhibitors and RT-PCR. HR efficiency and DNA damages were evaluated using Dr-GFP and comet assays. DNA morphology and DNA repair focus studies were analyzed using immunofluorescence. UALCAN portal was used to evaluate the expression of RAD51 and survival probability based on tumor stage, subtype, and race in breast cancer patients. **RESULTS:** Our results show that prexasertib treatment promotes both post-translational and transcriptional mediated regulation of BRCA1 and RAD51 proteins. Additionally, prexasertib-treated TNBC cells revealed over 55% reduction in HR efficiency compared to control cells. Based on these results, we hypothesized that prexasertib treatment induced homologous recombination deficiency (HRD) and thus should synergize with PARP inhibitors (PARPi) in TNBC cells. As predicted, combined treatment of prexasertib and PARPi olaparib increased DNA strand breaks, gammaH2AX foci, and nuclear disintegration relative to single-agent treatment. Further, the prexasertib and olaparib combination was synergistic in multiple TNBC cell lines, as indicated by combination index (CI) values. Analysis of TCGA data revealed elevated RAD51 expression in breast tumors compared to normal breast tissues, especially in TNBC subtype. Interestingly, there was a discrepancy in RAD51 expression in racial groups, with African-American and Asian breast cancer patients showing elevated RAD51 expression compared to Caucasian breast cancer patients. Consistent with these observations, African-American and Asian TNBC patients show decreased survival. **CONCLUSIONS:** Based on these data, RAD51 could be a biomarker for aggressive TNBC and for racial disparity in breast cancer. As positive correlation exists between RAD51 and CHEK1 expression in breast cancer, the in vitro preclinical data presented here provides additional mechanistic insights for further evaluation of the rational combination of prexasertib and olaparib for improved outcomes and reduced racial disparity in TNBC.

Pirotte, E. F., et al. (2018). "Sensitivity to inhibition of DNA repair by Olaparib in novel oropharyngeal cancer cell lines infected with Human Papillomavirus." *PLoS One* **13**(12): e0207934.

The incidence of Human Papillomavirus (HPV)-associated oropharyngeal squamous cell carcinoma (OPSCC) is increasing rapidly in the UK. Patients with HPV-positive OPSCC generally show superior clinical responses relative to HPV-negative patients. We hypothesised that these superior responses could be associated with defective repair of DNA double strand breaks (DSB). The study aimed to determine whether defective DNA repair could be associated with sensitivity to inhibition of DNA repair using the PARP inhibitor Olaparib. Sensitivity to Olaparib, and induction and repair of DNA damage, were assessed in a panel of 8 OPSCC cell-lines, including 2 novel HPV-positive lines. Effects on cell cycle distribution and levels of PARP1 and p53 were quantified. RNA-sequencing was used to assess differences in activity of DNA repair pathways. Two HPV-positive OPSCC lines were sensitive to Olaparib at potentially therapeutic doses (0.1-0.5 μ M). Two HPV-negative lines were sensitive at an intermediate dose. Four other lines, derived from HPV-positive and HPV-negative tumours, were resistant to PARP inhibition. Only one cell-line, UPCISCC90, showed results consistent with the original hypothesis i.e. that in HPV-positive cells, treatment with Olaparib would cause accumulation of DSB, resulting in cell cycle arrest. There was no evidence that HPV-positive tumours exhibit defective repair of DSB. However, the data suggest that a subset of OPSCC may be susceptible to PARP-inhibitor based therapy.

Sun, Q., et al. (2018). "MicroRNA-139-5P inhibits human prostate cancer cell proliferation by targeting Notch1." *Oncol Lett* **16**(1): 793-800.

Despite an improvement in the efficacy of chemotherapeutic agents, the outcome of patients with prostate cancer remains poor. MicroRNA (miRNA/miR)-139 expression is often downregulated in multiple types of tumor, including in prostate cancer. The aim of the present study was to investigate the inhibitory effect of miR-139 on the PC-3, C4-2B and LNCaP prostate cancer cell lines. Analysis of the cell cycle of PC-3, C4-2B and LNCaP cells transfected with miR-139 revealed a significantly increased percentage of cells in the G1 phase and a decreased percentage in the S and G2 phases compared with those transfected with a negative control miRNA. The growth inhibitory rate of miR-139-transfected cells 24, 48 and 72 h after transfection were 32.83 \pm 2.61, 52.58 \pm 3.2 and 62.36 \pm 4.55% in PC-3 cells; 30.28 \pm 2.25, 51.74 \pm 3.27 and 60.80 \pm 3.58% in C4-2B cells; and 33.20 \pm 2.67, 51.83 \pm 3.59 and 61.79 \pm 4.85% in LNCaP cells, respectively. The present study revealed that miR-139 inhibited the proliferation of prostate cancer cells by interfering with the cell cycle. Further study into the mechanism by which this happened suggested that miR-139 reduced cyclin D1 expression and inhibited cell proliferation through targeting Notch1.

Wu, C., et al. (2021). "PARP and CDK4/6 inhibitor combination therapy induces apoptosis and suppresses neuroendocrine differentiation in prostate cancer." *Mol Cancer Ther*.

We analyzed the efficacy and mechanistic interactions of PARP inhibition (PARPi) (olaparib) and CDK4/6 inhibition (CDK4/6i) (palbociclib or abemaciclib) combination therapy in castration-resistant prostate cancer (CRPC) and neuroendocrine prostate cancer (NEPC) models. We demonstrated that combined olaparib (Ola) and palbociclib (Palbo) or abemaciclib (Abema) treatment resulted in synergistic suppression of the p-Rb1-E2F1 signaling axis at the transcriptional and post-translational levels, leading to disruption of cell cycle progression and inhibition of E2F1 gene targets, including genes involved in DDR signaling/damage repair, antiapoptotic BCL-2 family members (BCL-2 and MCL-2), CDK1, and neuroendocrine differentiation (NED) markers in vitro and in vivo. In addition, Ola+Palbo or Ola+Abema combination treatment resulted in significantly greater growth inhibition and apoptosis than either single agent alone. We further showed that PARPi and CDK4/6i combination treatment-induced CDK1 inhibition, suppressed p-S70-BCL-2 and increased caspase cleavage, while CDK1 overexpression effectively prevented the downregulation of p-S70-BCL-2 and largely rescued the combination treatment-induced cytotoxicity. Our study defines a novel combination treatment strategy for CRPC and NEPC and demonstrates that combination PARPi and CDK4/6i synergistically promotes suppression of the p-Rb1-E2F1 axis and E2F1 target genes, including CDK1 and NED proteins, leading to growth inhibition and increased apoptosis in vitro and in vivo. Taken together, our results provide a molecular rationale for PARPi and CDK4/6i combination therapy and reveal mechanism-based clinical trial opportunities for men with NEPC.

Yang, X., et al. (2015). "JF-305, a pancreatic cancer cell line is highly sensitive to the PARP inhibitor olaparib." *Oncol Lett* **9**(2): 757-761.

Poly(ADP-ribose) polymerase-1 (PARP-1) is a DNA nick sensor involved in the base excision repair (BER) pathway. Olaparib, a PARP inhibitor, has demonstrated antitumor activity in homologous recombination (HR)-deficient cancers. To extend this specific therapy to other types of carcinomas, a panel of 11 different cancer cells were screened in the present study. JF-305, a pancreatic cancer cell line of Chinese origin, demonstrated sensitivity to the PARP inhibitor 6(5H)-phenanthridinone. In the present study, 3 μ M olaparib conferred a cell survival rate of 25% following four days of treatment. The colony formation efficiency was 83% at 10 nM, and dropped to 12% at 1 μ M following seven days of treatment. Furthermore, olaparib induced cell cycle arrest in the S and G2/M phases prior to the initiation of apoptosis. Although the incidence of double-strand breaks (DSBs) was increased in the olaparib-treated JF-305 cells, the RAD51 foci were well formed at the sites of gamma-H2AX recruitment, indicating an activated HR mechanism.

Furthermore, tumor growth was reduced by 49.8% following 22 days of consecutive administration of 10 mg/kg olaparib in the JF-305 xenograft mouse model. In summary, the JF-305 cell line was sensitive to olaparib and provided a prospective model for the preclinical assessment of PARP inhibitors in the therapy of pancreatic cancer.

REVIEWERS' COMMENTS:

Reviewer #1 (Remarks to the Author):

The authors addressed most of the previous concerns. I do feel that the revised manuscript better reflects the scientific significance of the work with respect to the experimental data presented.

Reviewer #3 (Remarks to the Author):

The authors have been responsive to reviewer comments well and have supplied an improved manuscript.